# Distinct synaptic transfer functions in same-type photoreceptors

**Cornelius Schröder[1,2]\*, Jonathan Oesterle[1,2], Philipp Berens[1,2,3†], Takeshi Yoshimatsu[4†‡], Tom Baden[1,4†]\***

[1]Institute for Ophthalmic Research, University of Tübingen, Tübingen, Germany; [2]Center for Integrative Neuroscience, University of Tübingen, Tübingen, Germany; [3]Bernstein Center for Computational Neuroscience,Centre for Integrative Neuroscience, all: University of Tubingen, Tubingen, Germany; [4]School of Life Sciences,University of Sussex, Sussex, United Kingdom

**Abstract** Many sensory systems use ribbon-type synapses to transmit their signals to downstream circuits. The properties of this synaptic transfer fundamentally dictate which aspects in the original stimulus will be accentuated or suppressed, thereby partially defining the detection limits of the circuit. Accordingly, sensory neurons have evolved a wide variety of ribbon geometries and vesicle pool properties to best support their diverse functional requirements. However, the need for diverse synaptic functions does not only arise across neuron types, but also *within*. Here we show that UV-cones, a single type of photoreceptor of the larval zebrafish eye, exhibit striking differences in their synaptic ultrastructure and consequent calcium to glutamate transfer function depending on their location in the eye. We arrive at this conclusion by combining serial section electron microscopy and simultaneous 'dual-colour' two-photon imaging of calcium and glutamate signals from the same synapse in vivo. We further use the functional dataset to fit a cascade-like model of the ribbon synapse with different vesicle pool sizes, transfer rates, and other synaptic properties. Exploiting recent developments in simulation-based inference, we obtain full posterior estimates for the parameters and compare these across different retinal regions. The model enables us to extrapolate to new stimuli and to systematically investigate different response behaviours of various ribbon configurations. We also provide an interactive, easy-to-use version of this model as an online tool. Overall, we show that already on the synaptic level of single-neuron types there exist highly specialised mechanisms which are advantageous for the encoding of different visual features.

**\*For correspondence:**
cornelius.schroeder@uni-tuebingen.de (CS);
t.baden@sussex.ac.uk (TB)

†These authors contributed equally to this work

Present address: ‡The School of Life Sciences, University of Sussex, Brighton, United Kingdom

**Competing interests:** The authors declare that no competing interests exist.

## Introduction

Ribbon-type synapses feed high-bandwidth sensory signals into their postsynaptic networks (reviewed in, e.g. *Moser et al., 2020*; *Sterling and Matthews, 2005*). However, depending on the species, modality, or receptor type, the nature of this synaptic transfer can differ greatly. For example, auditory systems typically operate at higher frequencies than visual systems, and accordingly auditory inner hair cells tend to use 'faster' ribbon synapses compared to those of photoreceptors (*Baden et al., 2013a*; *Moser et al., 2020*). Moreover, amongst photoreceptors of the vertebrate eye, rods and cones tend to differ greatly in the way they use their ribbons (*Regus-Leidig and Brandstätter, 2012*; *Sterling and Matthews, 2005*). Rods generally have large ribbons that can dock many hundreds of vesicles at a time, concentrated at a single release site, to support focussed, low-noise transmission (e.g. *Hays et al., 2021*). In contrast, cones usually use multiple smaller ribbons, often positioned at different release sites in a single pedicle, to serve diverse postsynaptic circuits (e.g. *DeVries et al., 2006*; *Jackman et al., 2009*). The ribbon synapses in electrosensory organs of elasmobranchs take such ribbon tuning to the extreme (*Bellono et al., 2018*). For

example, sharks achieve high-amplitude pulsatile transmission required for predation by combining greatly elongated ribbons with an ion-channel composition that supports broad spiking. In contrast, skates drive their smaller ribbons using graded voltage signals to support low-amplitude, oscillatory transmission suitable for intraspecific communication. This suggests that ribbons and their associated molecular machinery are important structural tuning sites of synaptic function in many sensory systems.

However, the functional requirements of synaptic transmission do not only differ across neuron types, but also *within* (*Baden et al., 2013b*; *Franceschini et al., 1981*; *Sinha et al., 2017*; *Szatko et al., 2019*; *Yoshimatsu et al., 2020b*; *Zimmermann et al., 2018*). For example, in vision, different parts of the eye survey different parts of visual space, often with distinct distribution of light and visuo-ecological significance (reviewed in *Baden et al., 2020*; *Land and Nilsson, 2012*). Correspondingly, we hypothesised that within-type functional tuning of a sensory receptor neuron should also utilise the vast tuning potential of its ribbon.

We explored this idea in the model of larval zebrafish UV-cone photoreceptors, which exhibit profound structural, molecular, and circuit differences depending on their location in the eye (*Yoshimatsu et al., 2020b*). First, UV-cone density varies across retinal regions and peaks in the acute zone (AZ), and to a lesser extent also nasally (*Yoshimatsu et al., 2020b*; *Zimmermann et al., 2018*). Furthermore, UV-cones in the AZ combine an enlarged outer segment with molecular tuning of the phototransduction cascade, an elevated calcium baseline and strong feedback from horizontal cells to boost detection brighter-than-background stimuli. This likely supports visual prey capture of UV-bright water-borne microorganisms such as paramecia. In contrast, UV-cones in other parts of the eye preferentially respond to darker-than-background stimuli, which may serve silhouette detection of nearby objects against the backdrop of bright UV-scatter from the sun. Amongst themselves, non-AZ UV-cones further differ in additional aspects, including their absolute light sensitivity. Building on these earlier results, we here asked if and how the actual synaptic transfer differs amongst UV-cones across the eye.

First, we used electron microscopy to reveal eye-region-specific structural differences amongst UV-cone ribbons and their presynaptic distribution of vesicles. Next, we found kinetic differences in synaptic transfer using in vivo simultaneous dual-colour two-photon imaging of the same pedicles' presynaptic calcium and resultant release. We then tied these findings together in a biophysical model, which enables computationally exploring possible underlying biological mechanisms and sites of tuning within the release cascade. Finally, we generalised our findings into an online model of synaptic transfer from the ribbon that enables free control over all key parameters, including ribbon dimensions, their dynamics, and the behaviour of underlying calcium drive (available online at http://www.tinyurl.com/h3avl1ga).

## Results

### UV-cone ribbon geometry and vesicle distributions differ with eye position

To establish possible structural differences amongst larval zebrafish UV-cone ribbon synapses, we obtained volumetric electron microscopy datasets of the outer retina taken from three different regions: , Acute zone (AZ), nasal (N), and dorsal (D). For each region, we anatomically identified (*Yoshimatsu et al., 2020a*) and 3D-reconstructed UV-cone pedicles ($n_{AZ, N, D}$ = 6, 6, 6) including their full complement of ribbons and surrounding vesicles (*Figure 1a–c*, Materials and methods). This revealed that dorsal ribbons were smaller (*Figure 1d, e*) and less numerous (*Figure 1f*) compared to AZ or nasal ones. However, nasal UV-cones had the lowest vesicle density immediately adjacent to the ribbon (*Figure 1g, h*, *Figure 1—figure supplement 1*). In addition, further away from the ribbon, the vesicle density was lowest in dorsal UV-cones and highest in AZ UV-cones (*Figure 1h*, *Figure 1—figure supplement 1*). Although it is possible that the experimental procedure distorted the vesicle distribution slightly, it is unlikely that this effect is disproportionately prominent in one eye region compared to the others. Taken together, the overall complement between ribbon number, geometry, and vesicle distributions therefore markedly differed across the three regions of the eye (*Figure 1i*). We next asked if and how these structural differences may translate into differences in synaptic function.

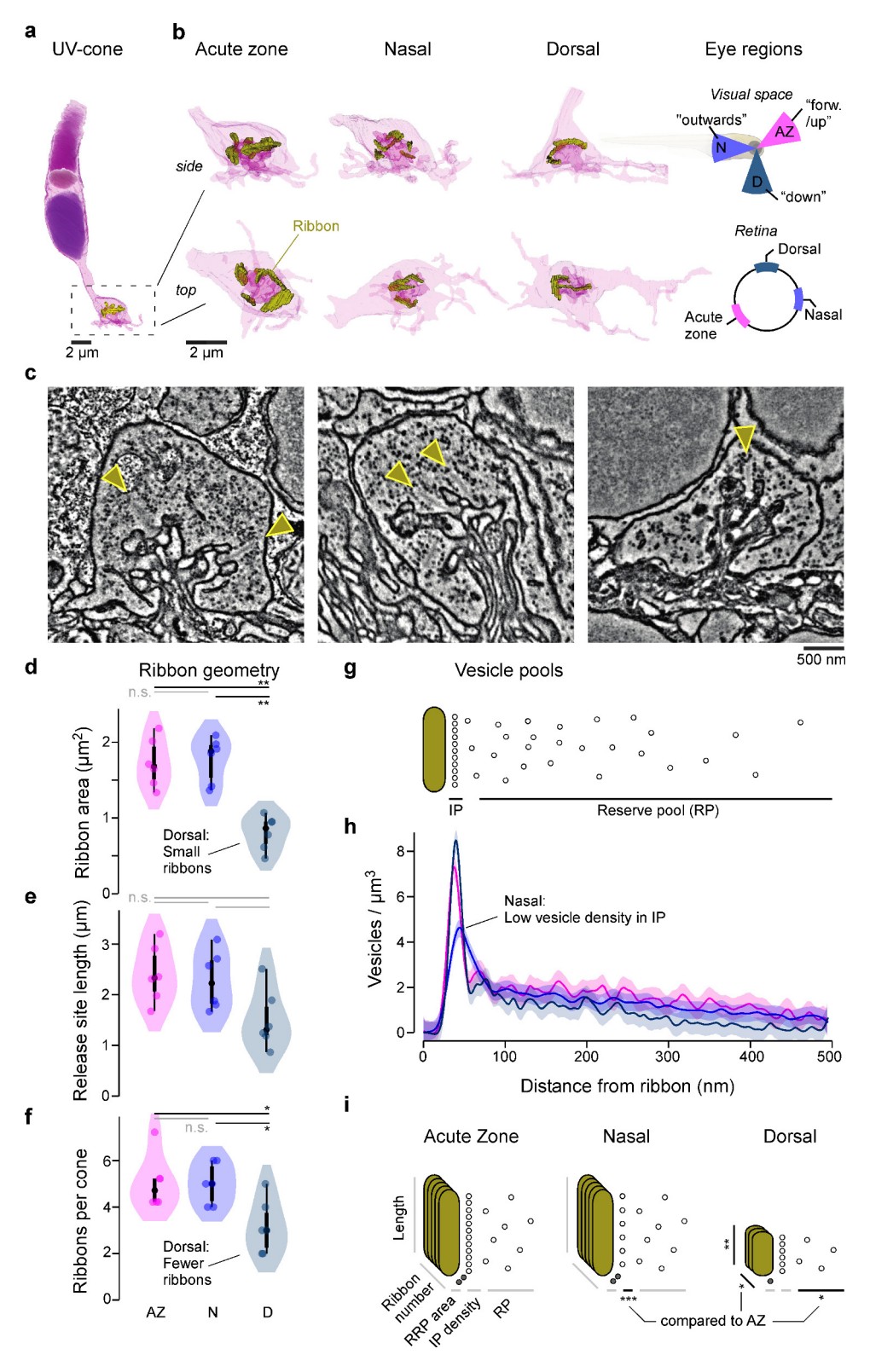

**Figure 1.** Eye-region-specific structural tuning of a ribbon synapse in UV-cones. (a) Example of a full UV-cone reconstruction, taken from the acute zone. Dark purple: nucleus; light purple: mitochondria; purple: outer segment; yellow: ribbon. (b) Zoom-ins of UV-cone terminals from different regions, which are illustrated on the rightmost panels. Ribbons are highlighted in yellow. Each terminal is shown from the side (top) and from below (bottom). (c) Example electron microscopy images from each zone, with arrowheads indicating ribbons. (d–f) Violin plots of the ribbon geometry and number for the
*Figure 1 continued on next page*

*Figure 1 continued*

three different regions (two-sided shuffling test with Bonferroni correction, $n_{AZ}$, $n_N$, $n_D$ = 6, 6, 6, *p<0.05, **p<0.01). (g) Two-dimensional schema of the vesicle pools at a ribbon synapse. (h) Mean and 95% confidence intervals for vesicle densities as a function of distance to the ribbon. Predictions are made from a generalised additive model (GAM, Materials and methods; see *Figure 1—figure supplement 1* for statistical comparisons). (i) Summary schema of observed Electron microscopic-level differences between UV-cones at the level of ribbon geometry, number, and vesicle distributions. Asterisks indicate significant differences compared to acute zone (AZ). The stacked ribbons (gold) indicate the ribbon number per cone, whereas the vesicles (small circles) are exemplified in a two-dimensional plane for a single ribbon.

The online version of this article includes the following figure supplement(s) for figure 1:

**Figure supplement 1.** Statistical comparison of spatial vesicle distributions.

## Eye-region-dependent differences in UV-cone release kinetics

To simultaneously monitor presynaptic calcium and resultant glutamate release from single UV-cone pedicles in vivo, we expressed the 'green' fluorescent glutamate biosensor SFiGluSnFR in horizontal cells postsynaptic to the cones, and the 'red' calcium biosensor SyjRGeco1b in all cones (*Figure 2a, b*, Materials and methods). Biosensor expression appeared to be uniform across the eye, in line with previous work (*Yoshimatsu et al., 2020b*). Furthermore, there was no obvious spectral mixing of the two fluorescence channels (*Figure 2—figure supplement 1a, b*). UV-cones were unambiguously identified based on their robust responses only to UV-light (Materials and methods; *Yoshimatsu et al., 2020a*). We then concurrently recorded red and green fluorescence signals under two-photon during presentation of 100% contrast widefield flashes of UV-light (3 s on, 3 s off), starting from a 50% contrast background (*Figure 2c*, Materials and methods). In example recordings from each eye region, this revealed different glutamate signals during light offsets, despite similar appearing calcium signals (*Figure 2c*). When scaled to their common sustained component, the dorsal cone glutamate release was much more transient compared to the nasal cone glutamate release, with a kinetically intermediate AZ cone (*Figure 2d*).

To systematically test for consistent differences between synaptic transfer in the three eye regions, we recorded paired calcium and glutamate signals from a total of n = 30, 16, 9 AZ, dorsal, and nasal UV-cones from n = 3, 4, 4 fish, respectively (*Figure 2—figure supplement 1c, d*). We then scaled and denoised each recording (Materials and methods) and computed the mean traces (*Figure 3a*) as well as key parameters relating to the amplitudes and kinetics of calcium and glutamate signals (*Figure 3b–j*; see also *Figure 3—figure supplement 1*). For this, all traces were scaled such that the UV-bright stimulus intervals had a zero mean and standard deviation of 1. We previously showed that these intervals correspond to the lowest possible calcium and glutamate release at the stimulus brightness used (*Yoshimatsu et al., 2020b*) and can therefore be used as a common baseline across zones. We present the rescaled traces (*Figure 3a*) in calcium units (c.u.) and vesicle units (v.u.), respectively, to be consistent with the used units in the model later. Based on the rescaled traces, we computed several indices of calcium and glutamate response amplitudes and their kinetics (*Figure 3b*), which together confirmed and extended initial observations from the single-pedicle examples (cf. *Figure 2*). In particular, both at the level of calcium (*Figure 3c*) and glutamate (*Figure 3d*), nasal UV-cones exhibited small peak amplitudes. In addition, the glutamate release of AZ UV-cones was increased during the 50% contrast period at the start of the stimulus (*Figure 3f*). In line with previous work, the AZ calcium baseline also appeared elevated (*Figure 3a*); however, this difference was not statistically significant (*Figure 3e*). This was likely related to the lower signal-to-noise ratio of jRGeco1b signals compared to those of GCaMP6f as used previously (*Yoshimatsu et al., 2020b*). Next, we quantified amplitudes of transient and sustained components at the level of glutamate (*Figure 3b, g–j*). For this, we analysed the first flash response separately from the mean of subsequent ones because all regions exhibited notably stronger adaptation during the first flash (*Figure 3a*). Overall, this consistently revealed the most pronounced within-pulse adaptation in dorsal UV-cones (*Figure 3g, h*), but no significant differences in the sustained components (*Figure 3i, j*).

To investigate the differences in presynaptic calcium in more detail, we reanalysed previously published data of calcium recordings with SyGCaMP6f in response to a 200 ms 'dark flash' (*Yoshimatsu et al., 2020b*). The kinetics of calcium responses were similar to each other across the three zones (*Figure 3k*), thus broadly supporting our previous results based on jRGeco

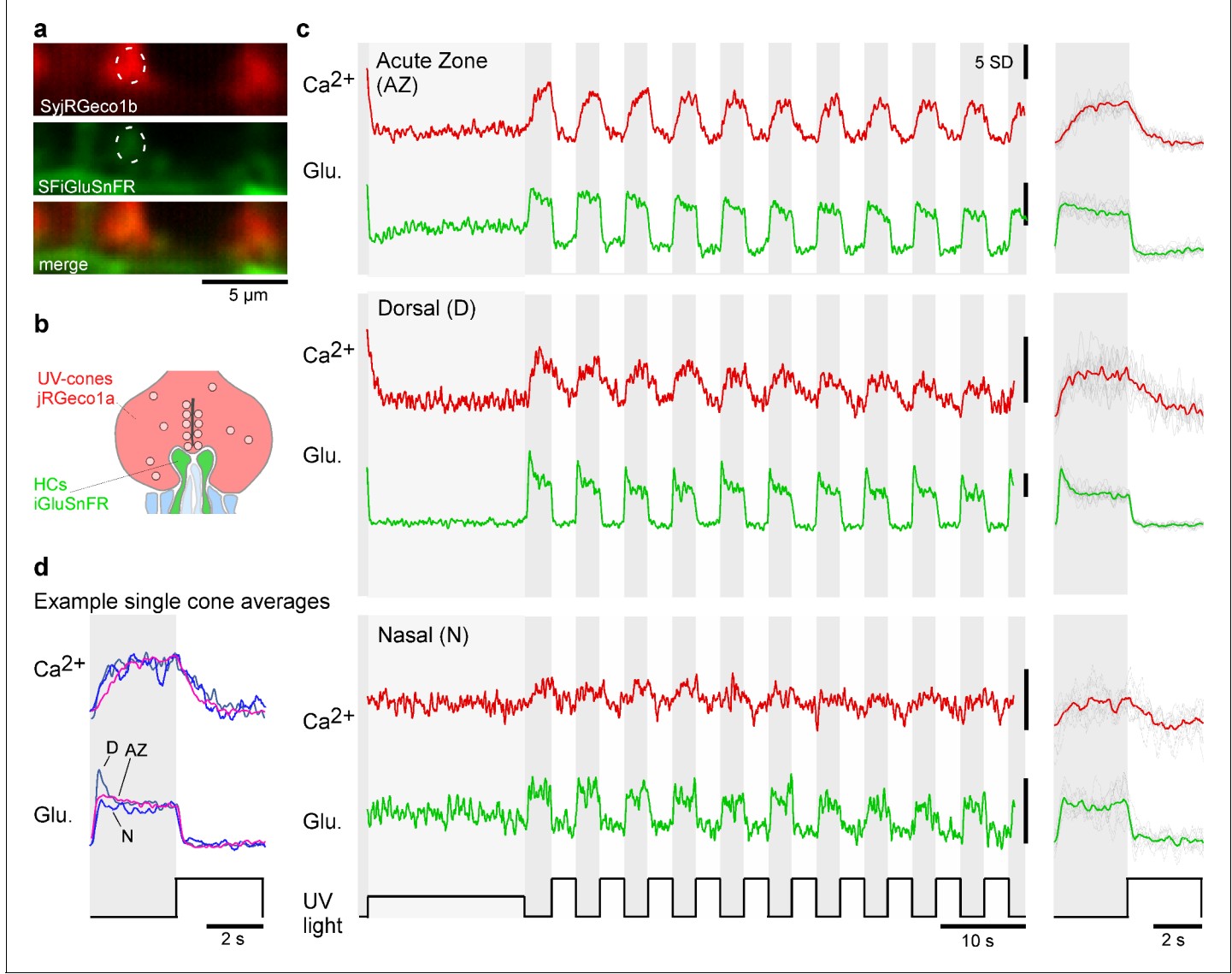

**Figure 2.** Simultaneous in vivo imaging of synaptic calcium and release. (a, b) Simultaneously acquired two-photon scans of cone terminals and opposing horizontal cell dendrites, with cone pedicles expressing SyjRGeco1b (red), and horizontal cell dendrites expressing SFiGluSnFR (green), and schematic representation, showing the cone pedicle (red) with ribbon and vesicles, as well as horizontal cell processes (green) and bipolar cell dendrites (blue). (c) Examples of raw calcium (red) and glutamate (green) traces recorded simultaneously from single UV-cones, one from each eye region as indicated. The averaged traces and superimposed stimulus repetitions are shown on the right. (d) Overlay of the averaged traces in (c), highlighting different glutamate responses despite very similar calcium responses.

The online version of this article includes the following figure supplement(s) for figure 2:

**Figure supplement 1.** Mean calcium and glutamate traces per region.

measurements. First, we found similar rise kinetics across all zones (*Figure 3l*) which therefore unlikely linked to the differences observed at the level of glutamate. Nevertheless, the response in the AZ was weakly but significantly advanced (i.e. it occurred earlier) compared to nasally or dorsally (*Figure 3n*). Moreover, decay kinetics were significantly faster in AZ cones compared to dorsal cones (*Figure 3m*) contrary to the adaptation index of the glutamate recordings (*Figure 3g, i*), hinting that the release dynamics were shaped differentially by the synaptic machinery across zones.

Taken together, our results so far highlight a range of structural (*Figure 1*) and functional (*Figures 2* and *3*) differences in the synaptic machinery of UV-cones across different regions of the eye.

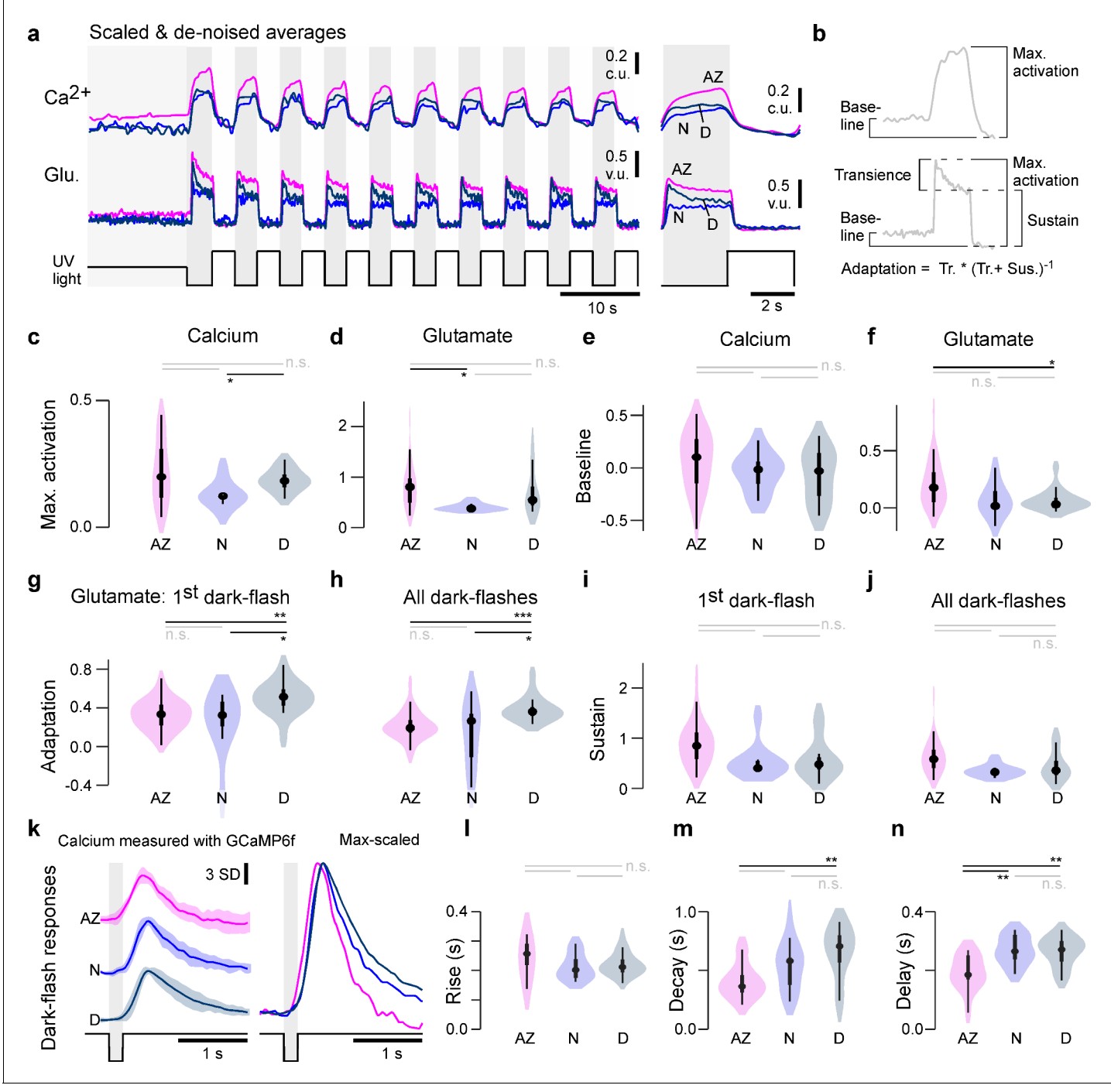

**Figure 3.** Physiological differences in light responses between UV-cones from different eye regions. (a) Scaled and denoised calcium and glutamate recordings averaged across multiple regions of interest (ROIs) (Materials and methods). We refer to the scaling as calcium units (c.u.) and vesicle units (v.u.) as the same traces also serve as input for the biophysical model (see *Figure 4*). (b) Schema of the calculated indices in (c–j) (Materials and methods). The transience index is computed as max.-sustainmax. (c–j) Quantification of physiological differences for the three different retinal regions (two-sided shuffling test with Bonferroni correction, $n_{AZ}$, $n_N$, $n_D$ = 30, 9, 16, *p<0.05, **p<0.01, ***p<0.001). (k) GCaMP6f recordings from *Yoshimatsu et al., 2020b*. Mean ± SD and overlaid mean traces in response to a 200 ms dark flash stimulus. (l–n) Quantification of physiological differences for the GCaMP6f recordings: time constants for an exponential rise, decay, as well as delay time to response (see also Materials and methods) (two-sided shuffling test with Bonferroni correction, $n_{AZ}$, $n_N$, $n_D$ = 13, 17, 22, *p<0.05, **p< 0.01, ***p<0.001).

The online version of this article includes the following figure supplement(s) for figure 3:

**Figure supplement 1.** Preprocessing of calcium and glutamate recordings.

## A model of glutamate release at the ribbon synapse

To systematically explore the possible mechanistic basis of the observed eye-wide variations in UV-cone synaptic functions, we next modelled the release machinery of the ribbon with a biophysical interpretable model that converts calcium signals to glutamate release (*Baden et al., 2014*; *Schröder et al., 2019*). The model consisted of three different vesicle pools (reserve pool [*RP*], intermediate pool [*IP*], and readily releasable pool [*RRP*]), the changing rates between these pools ($J^{RP\_IP}$, $J^{IP\_RRP}$, $J^{Exo}$), and a sigmoidal non-linearity with slope $k$ and offset $x_0$ which converts the calcium concentration into the final glutamate release (*Figure 4a*, Materials and methods). Building on recent advances in simulation-based Bayesian inference (*Gonçalves et al., 2020*; *Lueckmann et al., 2017*), we estimated posterior distributions over the model parameters for each of the region-specific datasets. In summary, the inference methods iterated the following steps over several rounds (*Figure 4b*): first, we draw samples from a prior distribution, evaluated the model, and extracted summary statistics on which the relevant loss was computed. Based on these loss values and the sampled parameters, a mixture of density network (MDN) was trained and finally evaluated to get the posterior of one round which was used as the prior for the next round. For the summary statistics, we defined features which captured all essential components of the release dynamics, such as transient and sustained components or peak heights (*Figure 4b*, Materials and methods). The model fitted the functional data well and accurately modelled the pronounced first UV-flash response differences between zones (*Figure 4c*). From here, the inferred posteriors (see *Figure 4—figure supplement 1a* for two-dimensional marginals) allowed us to compare the likely parameters between the different zones, including the estimated uncertainties. This allowed pinpointing possible key differences: for example, the calcium offset ($x_0$, which can be understood as the inverted calcium baseline; Materials and methods) was markedly increased in the nasal model (*Figure 4d, e*). In contrast, in case of maximal release rate, the posterior for the nasal data stayed close to the prior, indicating that this parameter was not essential to reproduce the traces. For RRP sizes and associated maximal refill rates, the model required the smallest values nasally (*Figure 4e*). The posteriors for IP size and associated refill rates were rather broad and did not allow to identify regional differences. Interestingly, with few exceptions (e.g. the maximal release rate and calcium offset), the parameter posteriors were mostly uncorrelated (*Figure 4—figure supplement 1a*), indicating that there is only little structure in the optimal parameter landscape. This also suggests that the model and data leave little room for possible compensatory mechanisms as this would result in a clear correlations of the involved parameters.

To confirm that the different model outputs do not simply rely on differences in the calcium inputs but rather on the differences of the inferred parameters, we compared the performance of the zone-specific models by shuffling the inferred parameters and glutamate datasets pairwise across zones (*Figure 4f, g*, *Figure 4—figure supplement 1b*). The match between the model output and measured glutamate release for a calcium input was generally worse for parameters corresponding to a different region, confirming that our release models were indeed regionally specific. For example, the sustained component of the AZ could not be captured by either the dorsal or nasal models (*Figure 4g*). Moreover, the models produced different transient behaviours independent of the calcium inputs (*Figure 4f, g*). A quantification of this model comparison shows that the shuffled models achieved lower loss values when evaluated on self- compared to non-self calcium inputs (*Figure 4—figure supplement 1b, c*). The model differences were especially high on the relevant loss (*Figure 4—figure supplement 1b*), which was based on the summary statistics and paid special attention to features like transiency. However, already the mean squared error (MSE) as loss function confirmed this difference: the region-specific models evaluated on self calcium inputs outperformed the models on non-self inputs (*Figure 4—figure supplement 1c*). Additionally, we compared the biophysical model to a statistical linear baseline model (Materials and methods). While as expected the linear model captured the general shape of the flash responses, it was not able to model glutamate transients nor adaptation over several flashes (*Figure 4h*). This also resulted in much higher loss values for the relevant loss function compared to the best biophysical model, which indicates the mismatch for essential features such as transiency (*Figure 4—figure supplement 1b*). The biophysical model also outperformed the linear model in terms of the MSE (*Figure 4—figure supplement 1c*). Together, this indicates that our modelling approach was sufficiently detailed for the posed problem, suggesting that the posterior distributions of each regional model can usefully

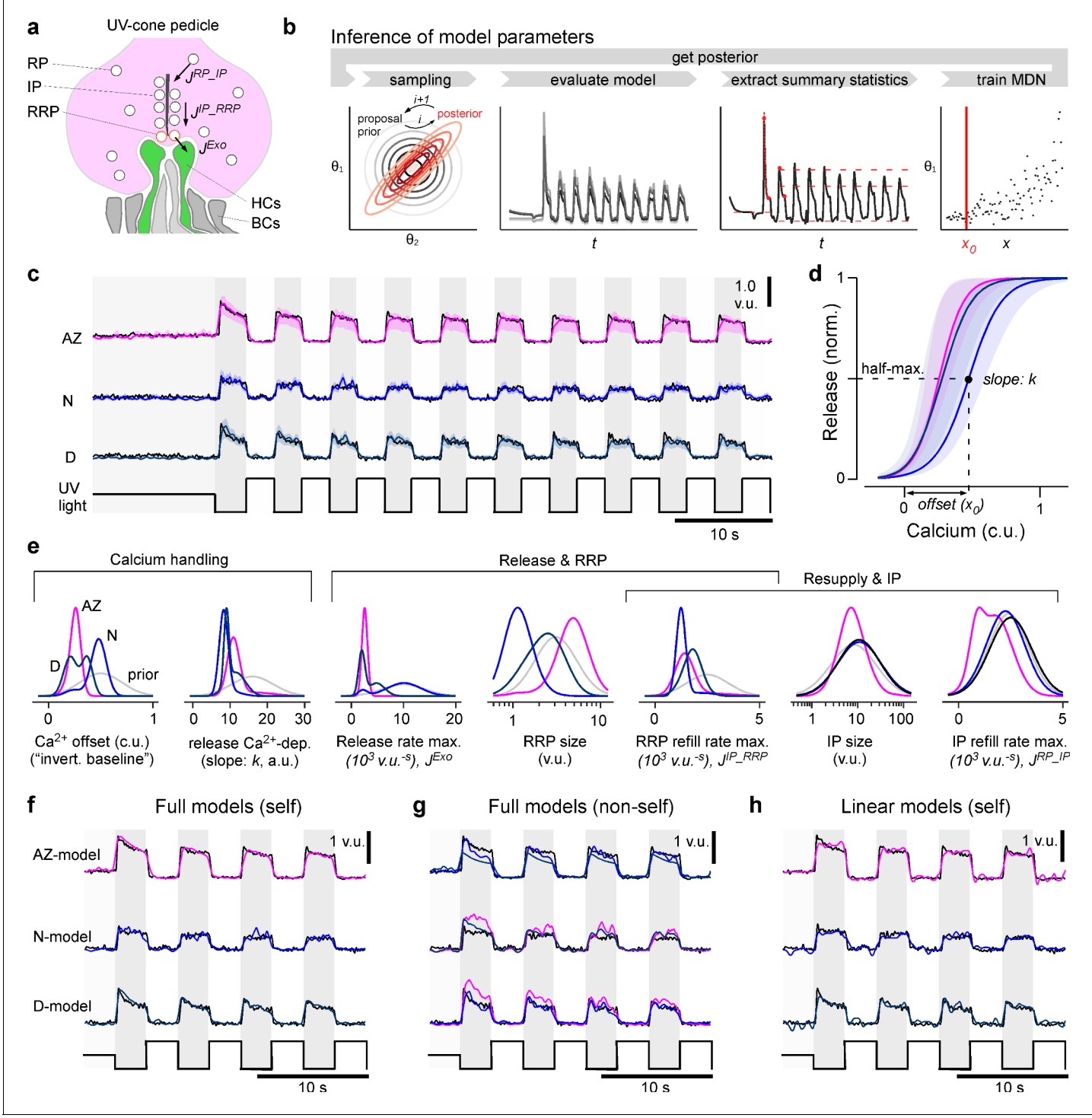

**Figure 4.** A model of calcium-evoked release from the ribbon. (a) Schema of the movement of vesicles at a ribbon synapse. The vesicles move from the reserve pool (*RP*) to the intermediate pool (*IP*) at the ribbon and finally to the readily releasable pool (*RRP*) close to the membrane before they are released into the synaptic cleft to activate the dendrites of invaginating horizontal cells (HCs). (b) Schema of the parameter inference: over several rounds, samples are first drawn from a prior, the model is then evaluated and summary statistics on which the relevant loss function is calculated are extracted. Based on these values and the sampled parameters, a mixture of density network (MDN) is trained and evaluated to get the posterior/prior for the next round. (c) Model predictions for the three regions (mode and 90% prediction intervals from posterior samples in colour, data in black). (d) Mode and 90% prediction intervals of the model's non-linearity. (e) Prior and one-dimensional marginals of the posterior distributions (see *Figure 4— figure supplement 1a* for the two-dimensional marginals). The units are vesicle units (v.u.) and calcium units (c.u.), referring to the scale invariance of the model. (f) Evaluation of the best region-specific models on its region-specific calcium traces. (g) Evaluation of best region-specific model on the

*Figure 4 continued on next page*

*Figure 4 continued*

calcium races of the other regions. (**h**) Evaluation of the linear baseline model (model in colour, data in black): especially the transient components are missed. See *Figure 3—figure supplement 1b* for a quantification of the goodness of fits.

The online version of this article includes the following figure supplement(s) for figure 4:

**Figure supplement 1.** Marginals of model posteriors, and comparison in model performance.

inform about the differences that underpin region-specific transfer functions from synaptic calcium to release via the ribbon.

## Predicting region-specific processes

An important strength of our modelling approach was that it allowed systematically exploring the possible influence of parameters such as vesicle pool sizes, their vesicle movements, and their calcium dependence, on the model output. To this end, we conducted a sensitivity analysis by computing the first-order Sobol indices (Materials and methods), a measure of the direct effect of each parameter on the variance of the model output. More specifically, it denotes the expected reduction in relative variance of the model output if we fix one parameter. For the computation of the Sobol indices, broadly speaking, a large number of parameters were drawn from the posterior distribution and the model was evaluated on these parameters. Afterwards the reduction in variance of the model evaluations was computed if one dimension of the parameter space was fixed (for details, see Materials and methods).The Sobol indices revealed a generally high influence of the calcium parameters ($x_0$ and $k$) during UV-bright periods and around light-dark transitions in all three zones (*Figure 5a–c*, top row). In contrast, beyond an initial key role of the RRP size for shaping the first dark-flash response, pool sizes generally played only relatively minor roles (*Figure 5*, bottom row). Instead, the most obvious region-wise differences occurred amongst vesicle transition rates between the pools. For example, the refilling rate of the IP from the RP ($J^{RP–IP}$) was increasingly critical for shaping dark flash responses in the AZ model (*Figure 5a*, blue) but had comparatively little influence in the nasal model (*Figure 5b*, blue). In contrast, the maximal release rate ($J^{Exo}$) particularly influenced the variance in early dark flash responses in the dorsal model (*Figure 5c*, green), while nasally refilling of the RRP from the IP ($J^{IP–RRP}$) played a greater role (*Figure 5b*, yellow). Together, this analysis suggests that particularly the rates of vesicle transfer between pools, rather than the pool sizes themselves or their calcium dependence, may underpin the experimentally observed region-wise differences in release properties from zebrafish UV-cones in vivo.

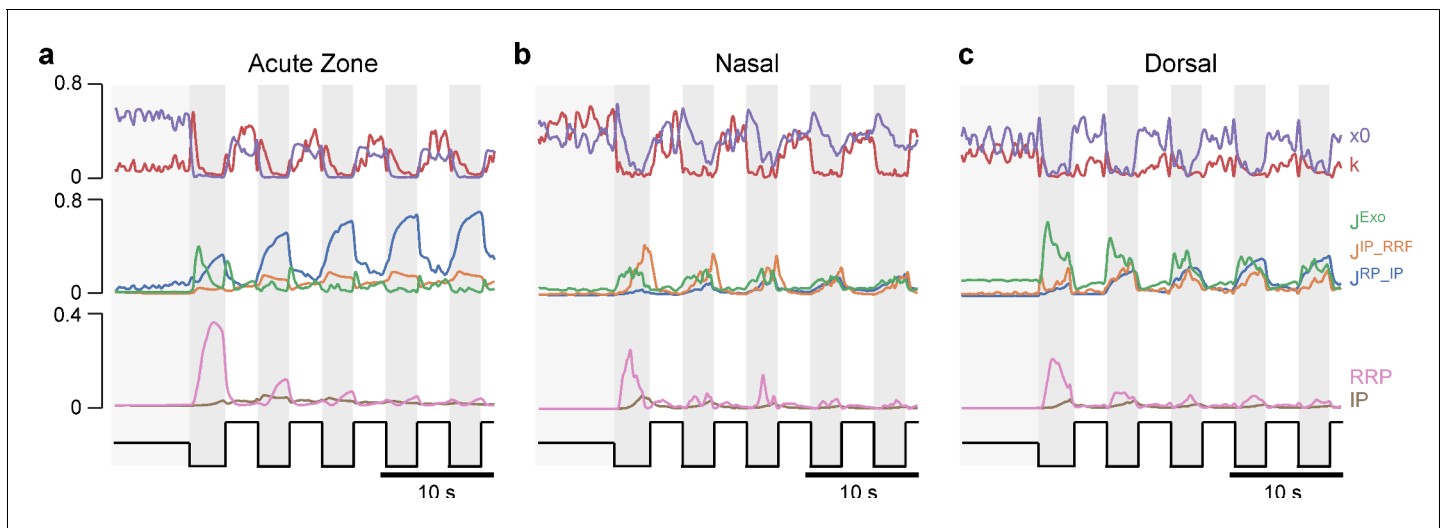

**Figure 5.** Sobol Indices. (**a-c**) Sensitivity of the model for the different model parameters measured by the Sobol index (Materials and methods). The Sobol index measures the expected reduction in relative variance for the fixation of parameter $\theta_i$. It depends on the posterior distribution and is therefore different for the fits to the three regions.

## General rules of ribbon tuning

We next sought to explore the general parameter landscape underlying release from the ribbon. For this, we calculated the same indices as in the corresponding data from 2P imaging (*Figure 3*), but this time on the model output obtained by simulating the model with parameters drawn from the posterior distributions. As expected, this reproduced the trends previously measured in vivo, including the low maximal activation nasally (*Figure 6a*), the largest transient component dorsally (*Figure 6b*), and the largest sustained component in the AZ (*Figure 6c*).

From here, we simplified the model by fixing the slope of the calcium non-linearity (*k*) and defining vesicle change rates as fractions of the corresponding pool sizes (Materials and methods). To be able to stimulate the model with arbitrary 'light' stimuli, we moreover implemented a linear calcium model based on a convolution with a biphasic kernel to reflect cone-activation by light (*Schnapf and Baylor, 1987*) and monophasic kernel to reflect calcium kinetics (*Baden et al., 2014*). This latter kernel was varied in subsequent simulations to explore the impact of calcium kinetics on synaptic performance (Materials and methods). Together, this allowed us to reduce the parameter space while at the same time identifying underlying computational principles. In the following, we always included the three fitted eye-region-specific parameter sets as a point of comparison (coloured 'dots' on top of heatmaps in *Figures 6* and *7*). These dots should be treated with some caution since in the simplified model they do not necessarily match the original ones in every dimension.

Exploring this model (*Figure 6d–f*), we found that both maximal activation (*Figure 6d*) and the size of the sustained component (*Figure 6f*) could be tuned by varying RRP and/or IP pool sizes (left column), with negligible contributions from the maximal release rate or the calcium offset (right column). In contrast, the transient component primarily hinged on the maximal release rate (*Figure 6e*, right column), with more complex additional contributions from the interplay of vesicle pool sizes (left column). Accordingly, our generalised model suggests that transient and sustained responses can be defined by largely non-overlapping properties of the ribbon, possibly providing a powerful handle for their independent tuning. For further exploration, the full model is available online as an interactive tool (*Figure 6g*) via *google colab* (http://www.tinyurl.com/h3avl1ga) or on *github* (https://github.com/coschroeder/cone_ribbon; *Schröder, 2021*; copy archived at swh:1:dir:0419f1165114b55870d1fff6719e92ba18cd2b82).

## Frequency dependence and event detection

In a final step, we explored the model behaviour on new stimuli and investigated the influence of the model parameters on different coding properties of the synapse. First, we measured the detectability of a high-amplitude dark event amongst an otherwise noisy stimulus sequence (*Figure 7a*, Materials and methods). This highlighted the calcium offset ($x_0$) as a key parameter (*Figure 7b*). Once $x_0$ is set, additional benefits could be gained from increasing RRP-size but only small benefits from increasing IP size (*Figure 7c*), ideally in further combination with a high maximal release rate (*Figure 7d*). We next measured the detectability of light events in the same way (*Figure 7e*, Materials and methods). This showed that beyond an inverse dependence on calcium baseline (*Figure 7f*, cf. *Figure 7b*), the ribbon parameters that benefitted the detection of on- and off-events were in fact virtually identical (*Figure 7g, h*, cf. *Figure 7c, d*). The low calcium offset (i.e. high baseline) supporting the detection of on-events is in line with our previous work (*Yoshimatsu et al., 2020b*), where the AZ showed highest calcium baseline and an enhanced ability to detect visuoecologically important UV-on events such as the presence of prey.

Finally, we explored how well different ribbon models could transmit fast temporal flicker, here summarised by a high-frequency index (HF*i*) (*Figure 7h*, Materials and methods). This revealed that this property primarily depended on the calcium kinetics rather than the specific tuning of the ribbon itself (*Figure 7i–k*). Beyond calcium dynamics (here, fixing the time constant $\tau$ for calcium at 0.5 s), again the calcium offset (*Figure 7j*) as well as an approximate balance of medium-sized pools for both the RRP and IP (*Figure 7k*), could provide additional support for encoding high-frequency components.

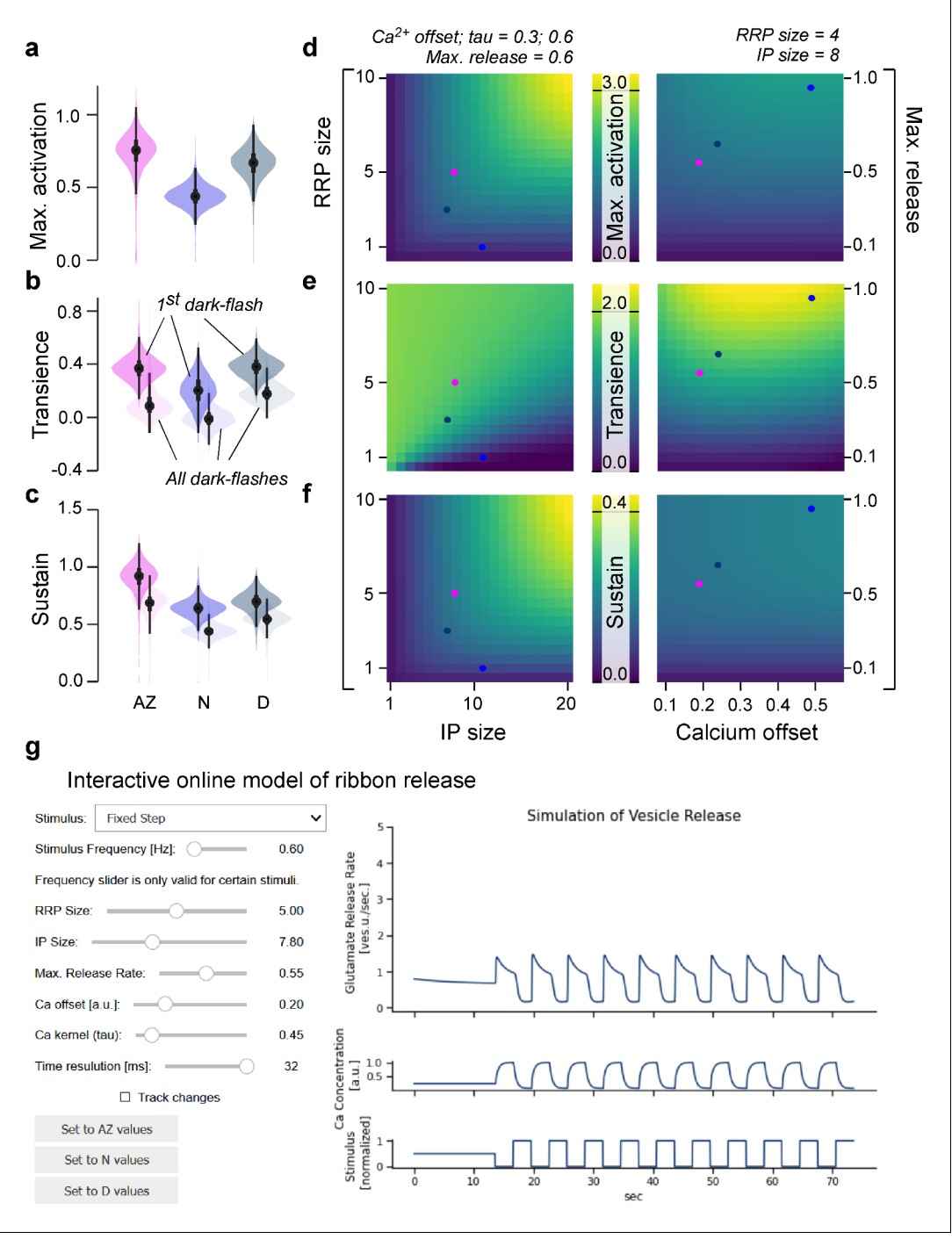

**Figure 6.** General rules of ribbon tuning: basic response parameters. (a-c) The same indices as in *Figure 3*, but here calculated on 10,000 model evaluations on samples from the posteriors. The model has learned the differences between the retinal regions and reproduces these differences (see *Figure 3* for comparison). (d-f) Indices as in (a–c), calculated on different parameter combinations as indicated. For this analysis, a step-stimulus feeding into a linear calcium model was added as the input to the release model (Materials and methods). For definition of the indices, see also *Figure 3b*. (g) Screenshot of the interactive online model, available at http://www.tinyurl.com/h3avl1ga.

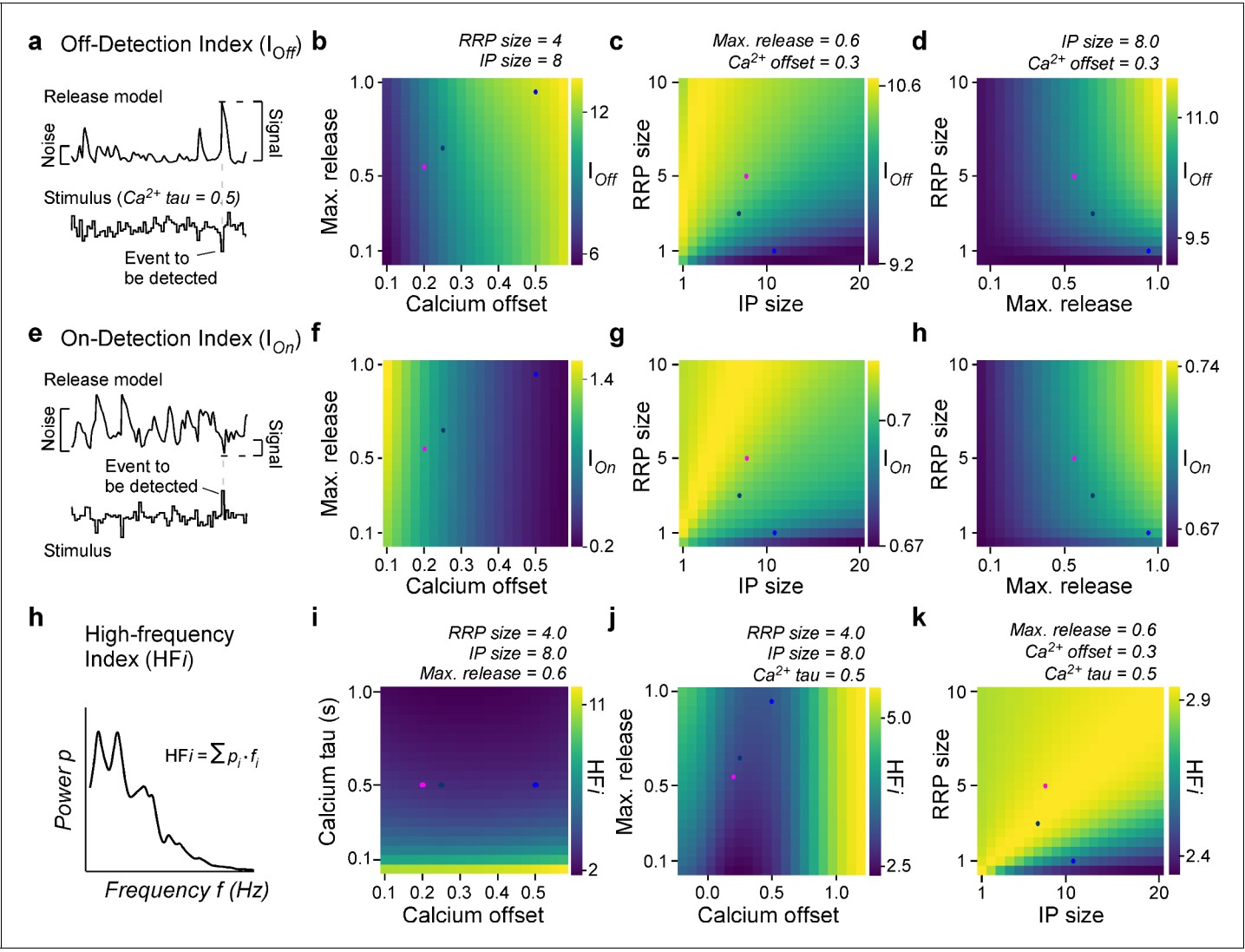

**Figure 7.** General rules of ribbon tuning: event detection and high-frequency encoding. (a) An off-detection index ($I_{Off}$) measures the model's baseline noise-normalised response amplitude to an off-event in the stimulus as indicated (Materials and methods). (b–d) $I_{Off}$ for different parameter combinations. (e–h) As (a–d), but for an on-detection index ($I_{On}$). (h) The high-frequency index (HF$i$) is a weighted sum of the discretised power spectrum and indicates the behaviour in a high-frequency regime. (i–k) HF$i$ for different parameter combinations. The fixed parameters are shown as titles in each panel. Note the different colour scales between panels. Further note that in (j) we explored the $x_0$ parameter space up to extreme response behaviour, which may not be physiologically plausible.

## Discussion

Combining ultrastructural evidence (*Figure 1*), in vivo dual-colour two-photon imaging (*Figures 2* and *3*) and computational modelling (*Figures 4* and *5*), we have shown how ribbon synapses belonging to the same neuron type can be regionally tuned to support distinct synaptic transfer functions depending on their location in the eye. Our findings complement and extend our recent demonstration that also upstream properties of these UV-cones are regionally tuned to support different visuoecological functions (*Yoshimatsu et al., 2020b*). We then further generalised this model to explore how specific properties of ribbon function can be principally used and traded off against one another to achieve a broad range of synaptic properties (*Figures 6* and *7*).

### Linking ribbon structure and function

Our findings that both UV-cone ribbon ultrastructure (*Figure 1*) and their effective synaptic transfer (*Figures 2* and *3*) systematically differed between regions support the notion that these two sets of

properties are linked (*Regus-Leidig and Brandstätter, 2012*; *Sterling and Matthews, 2005*; *Wichmann and Moser, 2015*). However, building a direct bridge between them remains difficult. This difficulty is part related to the absence of a direct experimental link between our Electron microscopic (EM) and Two-photon (2P) datasets, which we did not attempt in view of its extreme technical challenges (*Holler et al., 2021*) that to our knowledge have not been overcome for any ribbon synapse. Moreover, there is often more than one possible interpretation between a finding from EM and its functional consequence. For example, the size of a ribbon is expected to be linked to the number of vesicles it can hold. However, this link presumes that ribbon-attached vesicle density is fixed, which may not be the case, for example, due to conceivable variations in available binding sites or amongst vesicle transfer rates between pools (e.g. *Figure 4*).

Such possible complexity is illustrated in case of UV-cone regional variations: for example, dorsal ribbons were by far the smallest and least numerous (*Figure 1d, f*). From here, it is tempting to speculate that therefore their effective IP size, and perhaps also their RRP size, might be comparatively small. However, dorsal UV-cones also had a particularly high vesicle density near the ribbon, while instead nasal UV-cones had markedly reduced vesicle density here despite their otherwise large ribbons (*Figure 1g*, *Figure 1—figure supplement 1*). This strongly suggests that looking at ribbon geometry alone may be insufficient to accurately predict effective pool sizes. Instead, joint consideration of ribbon area and vesicle density may be more informative as supported by our model, which produced a near-identical IP size and generally low RRP sizes for both nasal and dorsal UV-cones (*Figure 4e*). In agreement, AZ cones combined large ribbons with a high vesicle density near the ribbon, and in their case the model did predict the largest RRP size (though a similar IP size). Although the differences of RRP sizes which were inferred by the model did not match the anatomical numbers quantitatively, we were able to infer the trends of the differences in the anatomical structures from pure functional data. Notably, model IP sizes generally stayed near the prior (*Figure 4e*) and exhibited low Sobol indices (*Figure 5*), which suggests that this property was not critical for explaining the glutamate responses to our relatively slow and simple test stimulus which was necessitated by the generally low signal-to-noise-ratio (SNR) of jRGeco1b signals (*Figure 2*). It remains possible that regionally distinct IP-size estimates would emerge in response when fitted to responses to different stimuli.

Beyond ribbon sizes, our ultrastructural analysis further highlighted an elevated vesicle density further away from the ribbon in AZ UV-cones (*Figure 1g*, *Figure 1—figure supplement 1*), potentially indicative of an increased RP and/or an increased rate of IP refilling. However, the model produced only minor differences in IP refill rates between regions, but in the reversed order (*Figure 4e*), and for simplicity RP size was fixed because it did not strongly affect function within the tens of seconds timescales interrogated (*Baden et al., 2014*). Accordingly, a direct link between UV-cone RP vesicle density and function remains outstanding. Conceivably, this property could be helpful for supporting the particularly high total vesicle turnover in AZ cones (*Figure 3d, f, i*) at longer timescales.

At the level of measured function, dorsal cones stood out in that their release was particularly transient (*Figure 3a, g, h*) as would be suited for pulsatile, rather than continuous transmission. Their generally small but densely populated ribbons and small RP may be well suited to support this property. In possible agreement, the model predicted a faster RRP refill rate in dorsal cones, which might be useful to ensure the RRP is rapidly replenished after each pulse. Conversely, the effective absence of any transient component nasal UV-cones (*Figure 3a, g, h*) in the model resulted in the lowest RRP refill rate, which may appear counterintuitive, but paired with a small nasal RRP resulted in the sustained response behaviour observed in *Figure 4f*. Additionally, this may further link to the above-mentioned low IP occupancy in nasal UV-cones, as observed under EM (*Figure 1g*). Moreover, the nasal model simultaneously predicted the smallest RRP but the largest RRP release rate, which resulted in a situation where the very few vesicles ready for release were immediately dumped, thus preventing the build-up of vesicles that enable the transients of dorsal and AZ UV-cones. Importantly, most functional differences across retinal regions appeared at the level of glutamate, whereas only subtle differences in the dynamics were observed at the level of calcium. The faster decay in AZ (*Figure 3m*) is linked to horizontal cells as reported previously (*Yoshimatsu et al., 2020b*), and the temporally advanced calcium signals measured in the AZ by GCaMP6f (*Figure 3n*) cannot explain the differences in the glutamate dynamics. Moreover, while it remains possible that the iGluSnFR

recordings slightly distorted the time courses of glutamate signals, it is unlikely to strongly affect our conclusions because any such effects would apply equally to all measured eye regions.

Taken together, while it remains difficult to make definitive links between ribbon structure and function, a number of inferences can be drawn which may usefully inform our understanding of the ribbon's role in moulding synaptic transmission to specific needs.

In the future it will be interesting to explore to what extent the differential ribbon tunings described here for larvae are also a characteristic of UV-cones in adult zebrafish. Adults feature a crystalline cone mosaic (reviewed in *Baden, 2021*), meaning that numerical anisotropies in cones as they occur in larvae (*Zimmermann et al., 2018*) are not expected. However, this does not preclude the possibility that UV-cones have different properties across the retina. While adult zebrafish display a much broader array of visual behaviours compared to larvae, their visual ecology remains poorly explored, which makes it difficult to predict what UV-cone tunings – if any – might be expected.

## Model and simulation-based inference

The presented model (*Figure 4a*) is a modified version of the basic framework used in *Baden et al., 2014*. Here we extended the approach by combining simulation-based inference with simultaneous calcium and glutamate recordings to extensively expand the analysis of the model. Moreover, rather than modelling discrete vesicle movements as in *Schröder et al., 2019*, our continuous model enabled the use of available software toolboxes (Materials and methods). With the simulation-based inference approach (*Figure 4b*) we obtained full posterior distributions (*Figure 4e*) rather than point estimates which can lead to overconfident or incorrect conclusions in the case of an under-constrained model or 'sloppy' model parameters. At the same time, posterior estimates can be seen as a global method to identify sloppy and stiff parameters (in contrast to local methods as in *Gutenkunst et al., 2007*) and additionally allowed us to conduct a sensitivity analysis by computing the first Sobol indices (*Figure 5*). These focus on the variance of the model output rather than on the variance of the posterior distributions and showed zone-dependent, biologically interpretable time courses. This highlighted a time-varying dependence of the release on the model parameters. It indicates that already a simple step stimulus is sufficient to show the influence of different anatomical properties on features of the signal. Finally, by establishing an interactive online model we encourage further exploration of the model and testing of hypotheses.

## General rules of ribbon tuning

Bringing together the model and observed differences in response behaviour, we tested our model on new stimuli and computed indices (*Figure 7*) which might be relevant to a broad range of sensory scenarios. For all investigated properties, we observed that the baseline and dynamics of the calcium signal were critical (*Figure 7b, f, i*), whereupon the anatomical ribbon properties allowed fine-tuning the final output behaviour (*Figure 7c, g, k*).

Interestingly, the influence of the calcium baseline for the detection of high-frequency events followed a bimodal distribution (*Figure 7j*) with intermediate values offering the poorest performance. In combination with the approximately equal and opposite effects of calcium baseline on the detectability of on- and off-events (*Figure 7b, f*), this suggests that the calcium baseline may present a key variable that enables ribbons to trade-off the transmission of high-frequency stimuli against providing an approximately balanced on- and off-response behaviour. Vice versa, it also suggests that the transmission of high-frequency events benefits from the use of a highly non-linear synapse that is either balanced to on- or off-events, but not both. This bimodal dependence of release performance on resting calcium levels is likely linked to biasing the system to preferentially either increase or decrease in light, but not both, thereby thresholding out responses of the respectively opposite polarity. Finally, and perhaps unsurprisingly, the time course of calcium decay was pivotal for defining the possible working range of high-frequency transmission regardless of the properties of the ribbon itself (*Figure 7i*, cf. *Figure 7j, k*). This effect was especially strong for a fast calcium decay (<100 ms) which is generally associated with nanodomains (*Jarsky et al., 2010*) rather than decay dynamics >>100 ms for microdomains (*Beaumont et al., 2005*). To what extent nano- or microdomain signalling dominates in larval zebrafish UV-cones remains untested. However, in view of their similarity in pedicle architecture to mammalian rods (e.g. small, single invagination site), it seems likely that also here already low micromolar calcium concentrations that are typically associated with

microdomains can evoke substantial release (*Thoreson et al., 2004*). In any case, a primary determinant for tuning a ribbon's high-frequency response is the local stimulus-driven calcium environment around the ribbon, rather than properties of the ribbon itself (see also *Baden et al., 2014*; *Baden et al., 2013a*). Nevertheless, once this is set, large pool sizes paired with a high release rate were generally preferable for all explored forms of signal detection. Taken together, our model therefore suggests that while the presynaptic calcium is a critical variable, even with a fixed calcium model the ribbon can be shifted into different response behaviours.

The effect and importance of calcium handling have also been shown in ribbon synapses of inner hair cells, where even within an individual synaptic compartment a local variation in calcium channels can lead to different transfer functions at different release sides (*Özçete and Moser, 2021*). Similar to the presented differences in UV-cones, such local heterogeneity in inner hair cells might help to diversify the sensory signal and highlight different features for downstream neurons.

# Materials and methods

## Animals

All procedures were performed in accordance with the UK Animals (Scientific Procedures) act 1986 and approved by the animal welfare committee of the University of Sussex. Animals were housed under a standard 14:10 day/night rhythm and fed three times a day. Animals were grown in 0.1 mM 1-phenyl-2-thiourea (Sigma, P7629) from 1 *dpf* to prevent melanogenesis. For two-photon in vivo imaging, zebrafish larvae were immobilised in 2% low melting point agarose (Fisher Scientific, BP1360-100), placed on a glass coverslip and submerged in fish water. Eye movements were prevented by injection of α-bungarotoxin (1 nl of 2 mg/ml; Tocris, Cat: 2133) into the ocular muscles behind the eye.

For all experiments, we used *6–7 days post fertilization* (*dpf*) zebrafish (*Danio rerio*) larvae. The following previously published transgenic lines were used: *Tg(cx55.5:nlsTrpR)*, and *Tg(tUAS:SFi-GluSnFR)* (*Yoshimatsu et al., 2020b*). In addition, *Tg(gnat2:SyjRGco1a)* lines were generated by injecting pTol2CG2-gnat2-SyjRGeco1a, plasmids into single-cell stage eggs. Injected fish were outcrossed with wild-type fish to screen for founders. Positive progenies were raised to establish transgenic lines.

The plasmid was made using the Gateway system (Thermo Fisher, 12538120) with combinations of entry and destination plasmids as follows: pDESTtol2CG2 (*Kwan et al., 2007*), p5E-gnat2 (*Lewis et al., 2010*; *Yoshimatsu et al., 2016*), pME-SyjRGeco1a, p3E-pA. Plasmid pME-SyjRGeco1a was generated by inserting a polymerase chain reaction (PCR)-amplified jRGeco1a (*Chen et al., 2013*; *Dana et al., 2016*) into pME plasmid and subsequently inserting a PCR-amplified zebrafish synaptophysin without stop codon at the 5′ end of jRGeco1a.

## EM

We used a previously published EM dataset of the larval zebrafish outer retina for this study (*Yoshimatsu et al., 2020a*). In the original paper, we only used one image stack from the AZ, but here we have in addition included two further stacks from nasal and dorsal regions, respectively. Image stacks were concatenated and aligned using TrackEM (NIH). The cones and ribbons were traced or painted using the tracing and painting tools in TrackEM2 (*Cardona et al., 2012*). Vesicle density on the ribbons was measured in six representative sections per cone. We selected sections where ribbons were aligned perpendicular to the sections. Ribbon release site length and area were measured in the 3D reconstruction of the ribbons.

## 2P imaging

All two-photon imaging was performed on a MOM-type 2-photon microscope (designed by W. Denk, MPI, Martinsried; purchased through Sutter Instruments/Science Products) equipped with a mode-locked Ti:Sapphire laser (Chameleon Vision-S, Coherent) tuned to 980 nm. We used two fluorescence detection channels for iGluRSnFR (F48x573, AHF/Chroma) and jRGeco1a (F39x628, AHF/Chroma), and a water immersion objective (W Plan-Apochromat 20×/1.0 DIC M27, Zeiss). For image acquisition, we used custom-written software (ScanM, by M. Mueller, MPI, Martinsried and T. Euler, CIN, Tuebingen) running under IGOR pro 6.3 for Windows (Wavemetrics). Recording configuration

was 124 × 32 pixels (2 ms per line, 15.6 Hz). Light stimuli were delivered through the objective by band-pass filtered light-emitting diodes (LEDs, 'red' 588 nm, B5B-434-TY, 13.5 cd, 8°, 20 mA; 'green' 477 nm, RLS-5B475-S; 3–4 cd, 15°, 20 mA; 'blue' 415 nm, VL415-5-15; 10–16 mW, 15°, 20 mA; 'ultraviolet, UV' 365 nm, LED365-06Z; 5.5 mW, 4°, 20 mA, Roithner, Germany).

All LEDs were jointly further filtered using FF01-370/36 (AHF/Chroma) and synchronized with the scan retrace at 500 Hz using a microcontroller as described in *Zimmermann et al., 2020*. The LED intensity was $1.10^5$ and $2.10^5$ photons per cone per second for adaptation period and UV flash, respectively, which corresponds to a low-photopic regime. A stimulus time marker embedded in the recording data was aligned to the traces with a temporal precision of 2 ms. For all experiments, animals were kept at constant background illumination for at least 5 s at the beginning of each recording to allow for adaptation to the laser. Regions of interest (ROIs), corresponding to individual presynaptic terminals of UV-cones, were defined manually. For calcium, we restricted ROIs within 1 µm from the release site at a terminal, while for glutamate we placed ROIs on the horizontal-cell dendrites immediately adjacent to a given cone, as previously (*Yoshimatsu et al., 2020b*). To unequivocally identify UV-cones, responses to the 'red', 'green', 'blue', and 'UV'-flashes were always recorded (always 1 s flash, 1 s darkness). Only cones that preferentially responded to the UV-LED were kept for further analysis (*Yoshimatsu et al., 2020a*).

## Scaling and denoising

We preprocessed the recorded and z-scored fluorescence traces as follows (see *Figure 3—figure supplement 1* for a visualisation): first, we applied a linear baseline correction to the calcium traces to correct a linear baseline decay. Then we rescaled the calcium and glutamate traces by z-scoring the traces with respect to the mean and standard deviation of the UV-bright stimulus intervals, resulting in a mean of 0 and a standard deviation of 1 in these intervals. By doing so we assumed that within these periods calcium channels are closed and recorded activity is either due to noise in the recording process or inherent channel/vesicle noise in the synapse. With this normalisation, we achieved a similar scaling for all traces independent of the level of indicator expression. Based on these preprocessed data (*Figure 3—figure supplement 1a, b*, second rows), the indices in *Figure 3c–j* were computed.

For the model input, we further averaged each zone over the trials and then applied a Butterworth filter of order 3 with a cutoff frequency of 5 Hz to denoise the signals. Since the output of the model was in vesicles per second, we finally shifted the processed glutamate data by its minimal value such that it had only positive values. To use the calcium concentration as input to the model, we additionally applied a Wiener deconvolution with the kernel of the calcium indicator JRGeco1 to the data (assuming a SNR of 10 for $f < 1$ Hz and a SNR of 1/20 for $f > 1$ Hz). The resulting pair of calcium/glutamate data per zone (*Figure 3—figure supplement 1*, last row, also *Figure 3a*) was finally used in the subsequent model.

## Analysis of the SyGCaMP6f data from *Yoshimatsu et al., 2020b*

The z-scored (GCaMP6f) calcium data was smoothed with a sliding average (window size of 100 ms). Then two exponential functions ($f(x) = c + a \cdot \exp(-\frac{1}{\tau} \cdot x)$) were fitted to the rise and decay periods. The time shift was defined via the start of the calcium rise, more precisely as $\min_{t}(Ca(t) > 3 \cdot \text{std}(Ca))$.

## Data indices

### Maximal activation, sustain, and transience

To be less prone to noise, we used the 90th and 50th percentiles ($p_i$) to calculate the maximal activation, transience, and sustain:

$$\text{max} = p_{90}\left(x_{[t_0, t_1]}\right),$$

$$\text{sustain} = p_{50}\left(x_{[t_2, t_3]}\right),$$

$$\text{transience} = \frac{\text{max} - \text{sustain}}{\text{max}},$$

where $x$ was the calcium/glutamate recording, respectively. The $t_i$ were chosen such that the first (for *max*) or the last (for *sustained*) second after the onset of the dark period is included.

## Off-, on-detection index ($I_{off}$, $I_{on}$), and high-frequency index (HFi)

For the detection indices, we used a Gaussian noise stimulus ($(\mu, \sigma) = (0.5, 0.3)$) with 2 Hz and a length of 150 s. This stimulus was shown for 120 s before the event to detect occurred with an amplitude of plus or minus four times the standard deviation and a duration of 500 ms. The detection indices of the simulation $x$ were finally computed as

$$I_{off}(x) = \left(\max\left(x_{[t_0, t_1]}\right) - \text{mean}\left(x_{[t_0, t_1]}\right)\right)/\text{std}\left(x_{[t_0, t_1]}\right),$$

$$I_{on}(x) = \text{abs}\left(\min\left(x_{[t_0, t_1]}\right) - \text{mean}\left(x_{[t_0, t_1]}\right)\right)/\text{std}\left(x_{[t_0, t_1]}\right),$$

with $t_0 = 60$ s and $t_1 = 150$ s.

For the HF*i* we used uniformly distributed noise of 100 s at 20 Hz and computed the discretised power spectrum $p$ of the simulation $x$ with Welch's method and defined the HF*i* by the standard deviation normalised data $x$ as

$$HFi(x) = \sum_{i=1}^{n} p_i(x) \cdot f_i,$$

with $n$ such that $f_n < 25$ Hz.

## Model

We modelled the synaptic release by a cascade-like ribbon synapse model (*Figure 4a*) with three vesicle pools (reserve pool *RP*, intermediate pool *IP*, and readily releasable pool *RRP*) and changing rates which were dependent on the occupancy of the pools (*Baden et al., 2014*; *Sterling and Matthews, 2005*). In this model, the glutamate release $e(t)$ was driven by the intracellular calcium $Ca(t)$:

$$e(t) = e_{max} \cdot f(Ca(t)) \cdot \frac{RRP(t)}{RRP_{max}}$$

with

$$f(Ca) = \frac{1}{1 + \exp(-k \cdot (Ca - x_0))}.$$

As $f(x_0) = 0.5x$, the parameter $x_0$ specifies the operating point of the non-linearity. It can be seen as an inverted baseline: the smaller $x_0$ the less additional calcium is needed to trigger a vesicle release. If we assume a fixed calcium affinity for vesicle release, this implies an increased baseline level in the synapse.

The changing rates $r(t)$ (between *RP* and *IP*) and $i(t)$ (between the *IP* and *RRP*) were independent of calcium:

$$r(t) = r_{max} \cdot \left(1 - \frac{IP(t)}{IP_{max}}\right) \cdot \frac{RP(t)}{RP_{max}}$$

$$i(t) = i_{max} \cdot \left(1 - \frac{RRP(t)}{RRP_{max}}\right) \cdot \frac{IP(t)}{IP_{max}}.$$

Additionally, the refilling $d(t)$ of the reserve pool was modelled by a constant factor of the available exocytosed vesicles $Exo(t)$:

$$d(t) = d_{max} \cdot Exo(t)$$

Therefore, the number of vesicles in the pools changed as

$$\frac{dRP(t)}{dt} = d(t) - r(t)$$

$$\frac{dIP(t)}{dt} = r(t) - i(t)$$

$$\frac{dRRP(t)}{dt} = i(t) - e(t)$$

$$\frac{dExo(t)}{dt} = e(t) - d(t).$$

In the original model in *Baden et al., 2014*, there was also a non-linear influence of calcium for $i$ ($t$) in terms of $\frac{Ca}{Ca+c}$. But initial runs of the fitting procedure resulted in $c \approx 0$ which indicated no calcium dependency for the changing rate $i$, and we excluded this term in the presented model. Since the endocytosis constant $d_{max}$ and the size of the reserve pool ($RP$) did not affect the output of the model, our fitting procedure was constrained to the remaining seven parameters: the changing rates ($r_{max}$, $i_{max}$, $e_{max}$), the non-linearity parameters ($k$ and $x_0$), and the remaining pool sizes ($IP$ and $RRP$). We call these parameters $\theta = (r_{max}, i_{max}, e_{max}, k, x_0, IP, RRP)$. From the equation above, it follows that the model is scale invariant: a scaling of the parameters (except the parameters for the non-linearity, $x_0$ and $k$) results in a scaled model output and thus only an arbitrary scale in v.u. to the experimental traces can be fitted.

The described coupled ODE was solved with *scipy*'s (version 1.5.1) implementation of the Bogacki–Shampine method (*Bogacki and Shampine, 1989*), an explicit Runge–Kutta method of order 3 with adaptive step sizes, where the maximal step size was set to the step size of the (calcium) input signal.

## Simplified model

To reduce the parameter space to identify general rules or ribbon tuning (*Figure 6d–f*, *Figure 7*, and online tool), we fixed $k$ to 10.2 (equal to the mean of the fitted parameters across zones) and additionally coupled the maximal changing rates to the pool sizes as follows:

$$r_{max} = 0.2 \cdot IP_{max}$$

$$i_{max} = 0.4 \cdot IP_{max}$$

$$e_{max} = \tilde{e}_{max} \cdot RRP_{max}$$

where $\tilde{e}_{max}$ can take values between 0 and 1.

For the general rules of ribbon tuning (*Figure 6d–f*, *Figure 7*, and online tool), the calcium concentration evoked by a light stimulus $s(t)$ is simulated as follows:

$$Ca(t) = \kappa_2 \sim * \sim \exp(\kappa_1 * s(t))$$

where $\kappa_1$ is a biphasic kernel from *Baden et al., 2014* and $\kappa_2$ is a double exponential kernel with fixed time constant $\tau_{rise}$ (30 ms) and variable decay parameter $\tau_{decay}$.

## Parameter inference

For the inference of the model parameters, we iterated the following steps over several rounds: first, we draw samples from a prior, evaluated the model, and extracted summary statistics on which the loss was computed. Based on these loss values and the sampled parameters, a MDN was trained and finally evaluated to get the posterior of this round which was used as the prior for the next round (*Figure 4b*).

### Summary statistics and relevant loss

A key ingredient in simulation-based Bayesian inference is to define problem- and domain-specific summary statistics to project the data to a low-dimensional feature space. For our setting, we identified the following 14 features $x_i$: baseline during adaption as well as period of UV-bright stimulus,

mean during UV-dark period as well as mean of the maximal release rates in this periods, maximal and minimal (as maximal value and 25th percentile) amplitude during the first flash, maximal and minimal (as maximal value and 25th percentile) amplitude of the second peak, and total number of released vesicles during first, second, and the last activation.

To pay special attention to the decay after the initial UV-dark flash, which we found to be informative about the different pool sizes of the ribbon, we fitted an exponential decay to this period. We used the inferred time constant $\tau$ and the evaluation of the exponential function at the time point before the next light onset as additional features along with an extra penalty if an exponential rise instead of a decay was fitted.

To calculate the relevant loss $R$ of a simulated trace $e$, we normalised the features in each component with the mean and standard deviation of the recorded traces. We then took a weighted MSE of this normalised summary vector $x$ to the normalised summary vector $x_0$ of the recorded trace as the relevant loss:

$$R(e) = \frac{1}{14} \sum_{i=1}^{14} w_i \left( x_{0,i} - x_i \right)^2.$$

The weights $w_i$ were not systematically optimised, but within reasonable ranges the results were relatively insensitive to the exact values. We chose $w = (0.5, 0.5, 5, 1, 1, 1, 1, 1, 1, 1, 2, 1, 1$ (0.01 for decay and $10 \cdot (1 + \text{ceil}(\tau))$ for rise)) for the features in the order mentioned as above, where the last value is the extra penalty for the exponential rise instead of a decay.

### Prior distribution and parameter normalisation

The modes of the prior were chosen as ($r$, $i$, $e$, $k$, $x_0$, $IP_{max}$, $RRP_{max}$) = (2.5, 2.5, 10, 14, 0.5, $\approx 13.8$, $\approx 4.0$), but as the model was scale invariant, the absolute values are uninformative and only the relative values are of interest. For technical reasons, we normalised the prior distributions such that the means of the uncorrelated multivariate normal distribution were 0.5 and the standard deviations were 0.2 in each dimension. For the IP and RRP pool sizes, we additionally exponentially scaled the sampled parameters, such that no negative values could occur.

### Parameter inference

We applied the Sequential Neural Posterior Estimation method described in *Lueckmann et al., 2017* (code available at https://github.com/mackelab/delfi) (also called SNPE-B) with some modifications which were also applied in *Oesterle et al., 2020*; *Yoshimatsu et al., 2020a*. In brief, SNPE-B draws over several rounds of samples $\{\theta i\}_{i \in I}$ from a prior $\tilde{p}(\theta)$ and evaluates the model for these parameters. For each evaluation $e_i$, the relevant loss function $x_i = R(e_i)$ is computed and a MDN $q_\phi(\theta, x)$ is trained on the data pairs $\{(\theta_i, x_i)\}_{i \in I}$. The posterior $p(\theta|x_0)$ is then calculated as $q_\phi(\theta|x = x_0)$ and used as a new prior $\tilde{p}(\theta)$ in the next sampling round. Instead of $x_0 = 0$, we calculated the new prior $\tilde{p}(\theta)$ in round $n$ as $q_\phi(\theta|x = \beta_n)$ where $\beta_n$ is the 0.1th percentile $\rho$ of the relevant loss function of all samples. It turned out that this is an efficient way to get a more stable behaviour of the MDN since it does not have to extrapolate to unreached loss values but is converging nevertheless. This evaluation of $q_\phi(\theta|x = \beta_n)$ at $\beta_n = \rho$ can be seen as the posterior over the parameters for the 'best-possible' model evaluations. Testing for different percentiles in a reasonable range did not change the results. We took the same approach for setting an adaptive bandwidth for the kernel. As an additional post-hoc verification of the posteriors, we took as final posterior distributions the posterior of the round with the smallest median loss of its samples ('early stopping').

### Technical details

We ran the inference algorithm over five rounds, with 300,000 samples per round. We chose three Gaussian components for the mixture of Gaussian distribution and a MDN with two hidden layers with 120 nodes each. In each round, the network was trained for 800 epochs with a minibatch size of 1000. To let the MDN focus on regions of low relevant loss values, we used a combined half-uniform-half-Gaussian kernel which was constant up to the pseudo observation $\beta_n$ and decayed then as a half-Gaussian. The scale of this half-Gaussian part of the kernel was in each round chosen as the 25 percentile of the relevant loss function.

## Sensitivity analysis

Sensitivity analysis was performed using *uncertainpy* (**Tennøe et al., 2018**) by using the *Gaussian-Mixture* probability distribution class of *chaospy* (**Feinberg and Langtangen, 2015**). Since model evaluations are computationally cheap and could run in parallel, we took the (quasi-) Monte Carlo method with 10e5 samples, resulting in 450,000 simulations. With uncertainpy we calculated the first-order Sobol sensitivity index $S_i$ which is defined as

$$S_i = \frac{\mathbb{V}[\mathbb{E}[Y|\theta_i]]}{\mathbb{V}[Y]},$$

where $\mathbb{E}$ and $\mathbb{V}$ are the expected value and the variance, respectively. $S_i$ measures the direct effect each parameter has on the variance of the model output (**Saltelli et al., 2008**; **Tennøe et al., 2018**). It tells us the expected reduction in relative variance if we fix parameter $\theta_i$. The sum of the first-order Sobol indices cannot exceed 1 and is equal to 1 if no interactions are present (**Glen and Isaacs, 2012**; **Tennøe et al., 2018**). We also calculated the total-order indices which gives the sensitivity due to interactions of the parameters. But since already the sum of the first-order indices is almost 1 for our model, the total order indices look quite similar and we omit their analysis.

It is important to remark that the Sobol indices are not a way to show how 'important' parameters are. A parameter which is overbearingly influencing the model would in the limit of the posterior estimation accumulate the mass on one single point. But this means that for samples from the posterior this parameter is not responsible for any variance of the model output and thus its Sobol index would be zero. The Sobol indices are therefore the result of a complex interplay of the model and the posterior estimation, but especially its temporal changes give us insight into the time-dependent influence on the model output. However, the posteriors are an adequate probability distribution to calculate the Sobol indices as they provide all parameter combinations which are in agreement with the experimental data and are more expressive as commonly used uniform distributions, where each marginal is simply defined as the mean ±10% of the fitted parameters.

## Linear baseline model

To evaluate the performance of the biophysical model, we compared it to a simple linear model. For this, we performed a regularised least square regression (ridge regression using scikit-learn, version 0.23.1, https://scikit-learn.org) to fit the calcium to glutamate response. In lack of a diverse enough dataset, we set the regularisation coefficient alpha to 0.1 and allowed the model to include the data from the past 0.5 s to predict the next time point.

## Statistical analysis

### Vesicle densities

We used generalised additive models (GAMs) for the comparison of the vesicle densities (**Figure 1g**, **Figure 1—figure supplement 1**). GAMs are an extension to generalised linear models by allowing linear predictors that depend on smooth functions of the underlying variables (**Wood, 2017**). We used the mgcv-package (version 1.8–33) in R on an Windows 10 workstation with default parameters, if not specified differently below. We modelled the dependence of the vesicle density as a smooth term dependent on the distance with 100 degrees of freedom and grouped by 'zone'. We further used 'zone' as additional predictive variable. The model explained ~65% of the deviance. Statistical significance for differences between the dependence on the vesicle density in the different retinal regions was obtained using the *plot_diff* function of the *itsadug*-package for R (version 2.4) with a 95% confidence level.

### Hypothesis testing

For comparisons between regional properties (ribbon geometry [**Figure 1**] and functional indices [**Figure 3**]), we used two-sided shuffling tests with Bonferroni correction. Sample sizes and significant levels are stated in the figure captions.

## Acknowledgements

The authors thank Leon Lagnado for critical feedback on the manuscript and Kit Longden for valuable discussions. Funding was provided by the European Research Council (ERC-StG 'NeuroVisEco' 677687 to TB), The Wellcome Trust (Investigator Award in Science 220277/Z/20/Z to TB), The UKRI (BBSRC, BB/R014817/1 to TB), the German Ministry for Education and Research (01GQ1601, 01IS18052C, 01IS18039A to PB), the German Research Foundation (BE5601/4-1, EXC 2064 – 390727645 to PB), the Leverhulme Trust (PLP-2017-005 to TB), and the Lister Institute for Preventive Medicine (to TB). Marie Curie Sklodowska Actions individual fellowship ('ColourFish' 748716 to TY) from the European Union's Horizon 2020 research and innovation programme. The authors also thank the FENS-KAVLI Network of Excellence and the EMBO YIP.

## Additional information

### Funding

| Funder | Grant reference number | Author |
|---|---|---|
| Wellcome Trust | 220277/Z/20/Z | Tom Baden |
| European Research Council | 677687 | Tom Baden |
| BBSRC | BB/R014817/1 | Tom Baden |
| Federal Ministry of Education and Research | 01GQ1601 | Philipp Berens |
| Federal Ministry of Education and Research | 01IS18052C | Philipp Berens |
| Federal Ministry of Education and Research | 01IS18039A | Philipp Berens |
| German Research Foundation | BE5601/4-1 EXC 2064 - 390727645 | Philipp Berens |
| Leverhulme Trust | PLP-2017-005 | Tom Baden |
| Lister Institute of Preventive Medicine | Fellowship | Tom Baden |
| H2020 Marie Skłodowska-Curie Actions | 748716 | Takeshi Yoshimatsu |

The funders had no role in study design, data collection and interpretation, or the decision to submit the work for publication.

### Author contributions

Cornelius Schröder, Conceptualization, Data curation, Software, Formal analysis, Validation, Investigation, Visualization, Methodology, Writing - original draft, Writing - review and editing; Jonathan Oesterle, Software, Methodology; Philipp Berens, Conceptualization, Supervision, Funding acquisition, Validation, Investigation, Project administration, Writing - review and editing; Takeshi Yoshimatsu, Conceptualization, Data curation, Formal analysis, Funding acquisition, Investigation, Methodology, Writing - review and editing; Tom Baden, Conceptualization, Resources, Supervision, Funding acquisition, Investigation, Visualization, Writing - original draft, Project administration, Writing - review and editing

### Author ORCIDs

Cornelius Schröder ![ORCID] https://orcid.org/0000-0001-5643-2097
Jonathan Oesterle ![ORCID] https://orcid.org/0000-0001-8919-1445
Philipp Berens ![ORCID] https://orcid.org/0000-0002-0199-4727
Takeshi Yoshimatsu ![ORCID] https://orcid.org/0000-0002-4939-2020
Tom Baden ![ORCID] https://orcid.org/0000-0003-2808-4210

## Ethics

Animal experimentation: All procedures were performed in accordance with the UK Animals (Scientific Procedures) act 1986 and approved by the animal welfare committee of the University of Sussex. All licensed procedures (in vivo 2-photon imaging of live zebrafish larvae) are covered by the Project License PPL PE08A2AD2 (to TB).

## Decision letter and Author response

Decision letter https://doi.org/10.7554/eLife.67851.sa1
Author response https://doi.org/10.7554/eLife.67851.sa2

# Additional files

## Supplementary files

- Transparent reporting form

## Data availability

Data is deposited at Dryad (https://doi.org/10.5061/dryad.7wm37pvt0).

The following dataset was generated:

| Author(s) | Year | Dataset title | Dataset URL | Database and Identifier |
|---|---|---|---|---|
| Schroder C | 2021 | Data from: Distinct Synaptic Transfer Functions in Same-Type Photoreceptors | https://doi.org/10.5061/dryad.7wm37pvt0 | Dryad Digital Repository, 10.5061/dryad.7wm37pvt0 |

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
