## [Decision Letter]

**Acceptance summary:**

This paper compares the properties of UV cone output synapses in different regions of the zebrafish retina using a combination of electron microscopy, quantitative imaging and computational modeling. These differences are related to ultrastructural differences in synaptic ribbons and evaluated using a previously-developed biophysical model for the operation of the synapse. The finding of regional differences in ribbon behavior is novel and suggests an under-appreciated degree of control of release by ribbon structure and behavior. A further nice feature of the work is that the model developed is made publicly available in an easy-to-use form.

**Decision letter after peer review:**

Thank you for submitting your article "Eye-Region Specific Ribbon Tuning Supports Distinct Modes of Synaptic Transmission in Same-Type Cone-Photoreceptors" for consideration by *eLife*. Your article has been reviewed by 3 peer reviewers, including Fred Rieke as Reviewing Editor and Reviewer #1, and has been overseen by Ronald Calabrese as the Senior Editor. The following individual involved in review of your submission has agreed to reveal their identity: Wei Li (Reviewer #3).

The reviewers have discussed their reviews with one another, and the Reviewing Editor has drafted this to help you prepare a revised submission. The reviewers agreed that the central question in the paper was broadly interesting and were impressed by the array of approaches used in the work. Several outstanding issues came up in review, however, that need to be strengthened before we can consider the paper further. All three reviewers agreed that these were necessary revisions.

Essential revisions:

1. Description of preprocessing steps. The imaging data went through several steps of processing before being compared across retinal regions. More controls are needed to build confidence that this initial processing did not introduce artifacts into the data – such as baseline shifts.

2. Calcium imaging data. The calcium measurements are noisy, and it is difficult to have confidence in the conclusion that calcium changes are similar across retinal regions. It would enhance the paper considerably if this data set could be strengthened.

3. Description of model. The connection between the model and the physiological data is hard to follow. It is important to clearly delineate what is learned from the model – including areas in which the model disagrees with experiment.

*Reviewer #1 (Recommendations for the authors):*

In a few cases non-statistically-significant trends in the data are noted (e.g. lines 206-207, 422-423). I have never found such statements helpful as it is not clear what to make of them. My preference would be to remove them.

It would be helpful to include axis labels on each plot – e.g. the probability distributions in Figure 4e.

Lines 288-290: this is not clear – can you explain how this works in a few extra sentences?

Line 386: what is "calcium T"? It seems quite long for a time constant defining the calcium dynamics.

*Reviewer #3 (Recommendations for the authors ):*

Specific questions and recommendations for the authors:

1. It will be helpful to have a retina diagram indicating the locations of three different regions.

2. Figure 1 d,e,f (and other figure panels in general) there is no need to mark n.s. On the other hand, in the Statistical Analysis section, GAMs models are mentioned only for Figure 1g, but not other results – needs a clarification.

3. Figure 1h is quite confusing, with a mixture of 3D and 2D plot, schematic drawing and statistical marks. What comparisons are these marks for? The legend is not specific and the Suppl Figure S1 doesn't clarify much.

4. It will be good to discuss the properties of the calcium sensor. Deconvolution of the calcium signal (lines 617-619) notwithstanding, presumably, the sensor has neither the temporal nor spatial resolution to catch the nano-domain calcium peak near the vesicles in RRP, which is critical for the release of RRP.

5. Likewise, the kinetics of iGluSnFR and of glutamate concentration in the cleft. Admittedly, Figures2a, 3c etc. show that the glutamate signal drops rapidly following the transition from dark to light, however, the rates of vesicle pool replenishment are a topic in the field-some discussion of how glutamate clearance from the cleft and the kinetics of the sensor will influence your estimates of replenishment rates would help future readers better interpret your findings in the context of their own observations.

6. In Figure 2d, the rising phase kinetics of the Glu for that nasal cone is strikingly different from that of the acute zone cone. However, such difference is not seen in Figure 3. Therefore, the one in Figure 2d may not be a good representation?

7. In Figure 3a, c.u. and v.u. (only defined in Figure 4 in the context of the model) were used here but not S.D. as in Figure 2, any explanation?

8. Lines 186-188, how were traces "normalized with respect to the UV-bright stimulus periods"?

9. Lines 194-195, "In addition, the glutamate release baseline of AZ UV-cones was increased during 50% contrast at the start of the stimulus" – it is unclear whether higher glutamate baseline occurred during the adaptation step (i.e. it increased during that period) or said increase was the level during adaptation compared to that during bright periods?

10. Lines 219-220, "a sigmoidal non-linearity with slope k and offset x0 which drives the final release" – this sentence is not clear, needs to clarify that it is referring to the relationship between calcium and release.

11. Lines 230-232, "x0 can be understood as the inverted calcium baseline (see Methods)" – Methods don't cover this point, though it is described in the f(Ca) equation, but it isn't obvious how x0 should be the inverted baseline, as if Ca=x0, f(Ca) = 0.5 (i.e., the point of half-release probability). Please clarify this. In general, there are places where explanations of model found in methods don't match those described in the main text (also see some of the points below). Please go over carefully to ensure consistency.

12. Figure 4e suggests a 5-10 times difference in RRP size between acute zone and nasal UV cones, which is not in line with the anatomical data (Figure 1h). Some discussions and clarifications will be helpful.

13. From Figure 4h, and Figure S3 b,c, the linear model doesn't look too bad (unless I misunderstand the figure panels, which are not explained in great detail). The explanation in lines 272-274 needs some work to make it clearer.

14. Sobol indices and their explanation are lacking. Are they computed using ca^2+^ and glutamate signals, or just glutamate? It is hard to parse their relative "contributions" to model behavior as described in the text, when the methods caution against interpreting this analysis as determining the "importance" of parameters (lines 805-806).

15. The sensitivity analysis suggests that vesicle transitions are more important than pool sizes or their calcium dependence. Thus, it appears that one intuition from the model is that ribbon size – the main anatomical difference of the UV cone ribbons from different regions – is not very important for the functional difference observed (also see discussion in lines 438-439). Although, it has been discussed that ribbon size does not necessarily correlate with IP or RRP size, but this appears to be the hallmark of the acute zone.

16. Lines 460-461, intuitively, a slower RRP refill rate will result in more transient response – after the depletion of RRP, less refilled vesicles to give the sustained component of the response. This is the opposite of what model predicted (a faster RRP). Some explanation and discussion will be helpful.

17. Also, the model simplifies vesicle transition rates by removing their calcium dependence. The Methods section indicates that this choice resulted from early fitting results that essentially "dialed out" the calcium dependence. Given the relative freedom that the model seems to have in finding suitable solutions, how is the lack of calcium dependence justified, and what potential impact might it have on the modeling results?

18. Lines 503-508, "In combination with the approximately equal and opposite effects of calcium baseline on the detectability of On- and Off-events (Figure 7b,f), this suggest(s) that the calcium baseline may present a key variable that enables ribbons to trade-off the transmission of high frequency stimuli against providing an approximately balanced On- and Off- response behaviour." – what will be the physiological relevance for such conditions, perhaps the level of adaptation? Any existing data or predictions?

19. I am slightly skeptical of the predictions that the model might make about the ribbon's frequency tuning (Figure 7) in light of the fact that the AZ model in particular seems unable to reliably capture the fast transient response to dark flashes (Figure 4 c,f).

---

## [Author Response]

Essential revisions:1. Description of preprocessing steps. The imaging data went through several steps of processing before being compared across retinal regions. More controls are needed to build confidence that this initial processing did not introduce artifacts into the data – such as baseline shifts.

We now include an additional supplementary Figure (Figure S3) and a more detailed description of the preprocessing steps to the methods section. The main differences in scale are arising by the z-scoring to the UV bright stimulus periods (meaning that the intervals corresponding to UV bright stimuli have mean of zero and a standard deviation of one). By this we assume a minimal activation during these periods, an assumption which we make explicit in the main text. Notably, the baseline shifts reported here are in line with the calcium baseline shifts demonstrated at some depth in our related previous work (Yoshimatsu et al. 2020, Neuron).

2. Calcium imaging data. The calcium measurements are noisy, and it is difficult to have confidence in the conclusion that calcium changes are similar across retinal regions. It would enhance the paper considerably if this data set could be strengthened.

We now reanalyse and include “new old” calcium imaging data from UV-cone pedicles (from Yoshimatsu et al., 2020, Neuron) recorded with SyGCaMP6f, which provides much higher signal to noise and displays faster kinetics. SyGCaMP6f is of course “green” and therefore cannot be simultaneously paired with “green” glutamate imaging. Nevertheless, we think that this data usefully adds to the characterisation of differences (and lack thereof) in the UV calcium response.

When measured with SyGCaMP6f, the kinetics of calcium responses to a 200 ms dark-flash are similar to each other across the three zones, thus broadly supporting our previous results based on jRGeco measurements. Nevertheless, the response in the acute zone (AZ) was weakly but significantly advanced (i.e. it occurred earlier) compared to nasally or dorsally. However, this had no measurable impact on the rise kinetics and was therefore unlikely linked to the differences observed at the level of glutamate. Finally, decay kinetics were significantly faster in AZ cones, which is linked to horizontal cells as reported previously (Yoshimatsu et al., 2020 Neuron).

We now include this additional data as part of the initial characterisation of calcium and glutamate responses across the three zones.

3. Description of model. The connection between the model and the physiological data is hard to follow. It is important to clearly delineate what is learned from the model – including areas in which the model disagrees with experiment.

We have now extended the discussion on the model results and discussed additional aspects. For example, we added a detailed discussion on how the nasal ribbon configuration is linked to its functional properties, or we improved the discussion on the Sobol indices.

A description of all changes can be found in the detailed response below.

Reviewer #1 (Recommendations for the authors):In a few cases non-statistically-significant trends in the data are noted (e.g. lines 206-207, 422-423). I have never found such statements helpful as it is not clear what to make of them. My preference would be to remove them.

Thank you, we have adapted the text.

It would be helpful to include axis labels on each plot – e.g. the probability distributions in Figure 4e.

It is common to show the marginal distributions (Figure 4e) without a scale bar since they can differ quite substantially for the different parameters and the exact scaling is less meaningful. See also for example here for similar visualization: Gonçalves, Pedro J., et al., "Training deep neural density estimators to identify mechanistic models of neural dynamics." *eLife* 9 (2020): e56261. We checked in detail the remaining Figures and added missing axis labels.

Lines 288-290: this is not clear – can you explain how this works in a few extra sentences?

The first order Sobol indices measure the direct effect of each parameter on the variance of the model output. More specifically, it tells the expected reduction in relative variance of the output if we fix one parameter. We added an explaining sentence, more details can be found in the Methods.

Line 386: what is "calcium T"? It seems quite long for a time constant defining the calcium dynamics.

This was a typo and was meant to be the time constant tau for the calcium decay. We fixed it in the manuscript. The approximate value of 0.5 sec was inferred from the calcium recordings, we omitted to show the kernel, but it is included in the online visualization. Also other time constants can be tested in there.

Reviewer #3 (Recommendations for the authors ):Specific questions and recommendations for the authors:1. It will be helpful to have a retina diagram indicating the locations of three different regions.

The requested diagram has been added to Figure 1.

2. Figure 1 d,e,f (and other figure panels in general) there is no need to mark n.s. On the other hand, in the Statistical Analysis section, GAMs models are mentioned only for Figure 1g, but not other results – needs a clarification.

We find the “n.s.” labels useful, in part because in some panels none of the differences were significant and the label makes this quite explicit. Accordingly, we have opted to retain them. GAMs were indeed only used for Figure 1g – this is motivated by the difference in data structure of this panel compared to others (i.e. a comparison between continuous rather than discrete distributions). We now clarified this in the methods and added a short paragraph on the used testing procedure.

3. Figure 1h is quite confusing, with a mixture of 3D and 2D plot, schematic drawing and statistical marks. What comparisons are these marks for? The legend is not specific and the Suppl Figure S1 doesn't clarify much.

The asterisks are meant to indicate a statistically significant difference in the indicated property (e.g. ribbon size/number) relative to the acute zone. We apologise for not making this clear in the previous version, it is now directly noted in the panel. Regarding the 2D/3D representation, we agree that it may be a little confusing, but we cannot think of a “better” way of summarising all properties analysed by EM in a single panel, so we opted to keep it. We did however expand on the related explanation in the legend to further clarify what is shown.

4. It will be good to discuss the properties of the calcium sensor. Deconvolution of the calcium signal (lines 617-619) notwithstanding, presumably, the sensor has neither the temporal nor spatial resolution to catch the nano-domain calcium peak near the vesicles in RRP, which is critical for the release of RRP.

This point seems to link to the ongoing debate on to what extent release from ribbons is driven by micro- and/or nano-domain calcium signalling. It is our understanding that this debate remains unresolved in a truly general sense. Rather, it seems to be non-mutually exclusive (i.e. both micro and nano-domain signals working together), and moreover quite specific to each ribbon synapse in question. In larval zebrafish cones, the pedicle has a rather small cytoplasmic volume, there is only one invagination from postsynaptic processes, and all ribbons inside the cone are opposed to this single invagination. Accordingly, on a possible “sliding scale” of micro- vs nano-domain dominance, we think it is likely that in larval zebrafish cones microdomains will have a notable impact on release. While we are not aware of any data directly looking at this question in zebrafish larval UV-cones, there is good data available from systems that are perhaps quite similar, such as mammalian rods (which also have a single invagination site). For example, Thoreson et al., 2004, Neuron, Figure 3.

5. Likewise, the kinetics of iGluSnFR and of glutamate concentration in the cleft. Admittedly, Figures2a, 3c etc. show that the glutamate signal drops rapidly following the transition from dark to light, however, the rates of vesicle pool replenishment are a topic in the field-some discussion of how glutamate clearance from the cleft and the kinetics of the sensor will influence your estimates of replenishment rates would help future readers better interpret your findings in the context of their own observations.

We agree that there are technical limitations as to what the iGluSnFR signal can tell us about the exact dynamics of glutamate in an unperturbed situation. Likely this will never be fully addressable. Rather, we use the iGluSnFR signals in a comparative fashion across eye regions, where presumably any distortion of the signals as alluded to by the reviewer would be approximately equal. Following the reviewer’s suggestion, we now explain this more directly in the main text.

6. In Figure 2d, the rising phase kinetics of the Glu for that nasal cone is strikingly different from that of the acute zone cone. However, such difference is not seen in Figure 3. Therefore, the one in Figure 2d may not be a good representation?

Thanks, we agree. We have replaced the nasal example with a more representative trace.

7. In Figure 3a, c.u. and v.u. (only defined in Figure 4 in the context of the model) were used here but not S.D. as in Figure 2, any explanation?

After scaling, SD adopts arbitrary units. For consistency with the model later we decided to use c.u. and v.u. Here (i.e. “calcium units”, and “vesicle units”). We agree that this could be explained better, and have now rephrased as follows:

“We show the rescaled traces in c.u. (calcium units) and v.u. (vesicle units) respectively, to be consistent with the used units in the model later.”

8. Lines 186-188, how were traces "normalized with respect to the UV-bright stimulus periods"?

The traces were rescaled such that the UV-bright stimulus periods had a mean of zero and a standard deviation of one. We included this missing piece of information and expanded additionally the explanation of the pre-processing.

9. Lines 194-195, "In addition, the glutamate release baseline of AZ UV-cones was increased during 50% contrast at the start of the stimulus" – it is unclear whether higher glutamate baseline occurred during the adaptation step (i.e. it increased during that period) or said increase was the level during adaptation compared to that during bright periods?

Thank you, we meant the former (i.e. glutamate release “is” higher during the adaptation step). This is now clarified in the text.

10. Lines 219-220, "a sigmoidal non-linearity with slope k and offset x0 which drives the final release" – this sentence is not clear, needs to clarify that it is referring to the relationship between calcium and release.

Thanks, this is now clarified in the manuscript.

11. Lines 230-232, "x0 can be understood as the inverted calcium baseline (see Methods)" – Methods don't cover this point, though it is described in the f(Ca) equation, but it isn't obvious how x0 should be the inverted baseline, as if Ca=x0, f(Ca) = 0.5 (i.e., the point of half-release probability). Please clarify this. In general, there are places where explanations of model found in methods don't match those described in the main text (also see some of the points below). Please go over carefully to ensure consistency.

x0 can be seen as an inverted baseline as it shifts the whole linearity to a different operating point: the smaller x0 the less additional calcium is needed to trigger vesicle release. If we assume a fixed calcium affinity this implies an increased baseline level.

We apologise for having omitted these explanations in the initial manuscript, we have expanded the explanation in the Methods of the revised manuscript.

12. Figure 4e suggests a 5-10 times difference in RRP size between acute zone and nasal UV cones, which is not in line with the anatomical data (Figure 1h). Some discussions and clarifications will be helpful.

As we note in the manuscript, it is difficult to quantitatively link anatomical structures to functional data. However, the small RRP size in the nasal zone inferred by the model (Figure 4e) matches very well to the low vesicle densities at a small distance from the ribbon in the nasal zone in Figure 1. Our model thus picks up the right trends for an anatomical structure from pure functional recordings, which is in our opinion already remarkable given the experimental noise and fine-grained differences.

We commented on this point in the revised manuscript.

13. From Figure 4h, and Figure S3 b,c, the linear model doesn't look too bad (unless I misunderstand the figure panels, which are not explained in great detail). The explanation in lines 272-274 needs some work to make it clearer.

Compared to the “best model”, the linear model clearly lacks in accuracy, perhaps most intuitively visible when looking at adaptation kinetics. This is especially the case for the relevant loss, which is based on the summary statistics. We extended the mentioned lines and hope to clarify it now in the manuscript.

14. Sobol indices and their explanation are lacking. Are they computed using Ca^2+^ and glutamate signals, or just glutamate? It is hard to parse their relative "contributions" to model behavior as described in the text, when the methods caution against interpreting this analysis as determining the "importance" of parameters (lines 805-806).

The first order Sobol indices measure the direct effect of each parameter on the variance of the model output. More specifically, it tells us the expected reduction in relative variance of the output if we fix one parameter. For the computation, broadly speaking, many parameters were drawn from the posterior distribution and the model was evaluated on these parameters. Afterwards the reduction in variance of the model evaluations was computed if one dimension of the parameter space was fixed.

We agree that they are non-intuitive to interpret for a single time point, however its temporal changes give us insight into the time dependent influence on the model output. Often Sobol indices are computed by drawing random samples from a uniform distribution on a high dimensional cuboid [r1,s1] x … x [rn,sn] where each interval [ri,si] is simply defined by the mean+-10% of the parameter fit, where the definition of 10% leaves much room for interpretation and could not be meaningful in the same way for all parameters. We believe that the inferred posterior distributions are a much better suited probability distributions as they encode all parameter combinations which agree with the experimental data.

We expanded our explanation on this point in the manuscript.

15. The sensitivity analysis suggests that vesicle transitions are more important than pool sizes or their calcium dependence. Thus, it appears that one intuition from the model is that ribbon size – the main anatomical difference of the UV cone ribbons from different regions – is not very important for the functional difference observed (also see discussion in lines 438-439). Although, it has been discussed that ribbon size does not necessarily correlate with IP or RRP size, but this appears to be the hallmark of the acute zone.

As the reviewer notes, one potentially interesting hint from our work is that ribbon size does not necessarily translate 1:1 to vesicle pool sizes, or their relative transition rates. One particularly clear example of this might come from comparing Figures 1d-f and Figure 1h, between nasal and acute zone. Both have similar ribbon geometry (Figure 1d-f), but nasal ribbons nevertheless appear to pack fewer vesicles (Figure 1h). Linking with our functional data and modelling, it then appears that perhaps on top of that, vesicles simply move at different rates between the pools, a property that is impossible to pick up from a static EM reconstruction.

More generally, as mentioned in the manuscript and discussed in the previous point, it is difficult to judge the overall importance of a parameter from the sensitivity analysis. However, we clearly see time dependent effects of the different parameters and especially the RRP size matters for the transient component, which can be seen in Figure 5. Indeed, the pattern for IP size seems to be different and it may be that case that the used stimulus is not optimal to infer this parameter from functional recordings.

How the ribbon size relates to different vesicle densities and how these densities could potentially influence the changing is however still an open question and cannot be answered in the scope of this manuscript.

16. Lines 460-461, intuitively, a slower RRP refill rate will result in more transient response – after the depletion of RRP, less refilled vesicles to give the sustained component of the response. This is the opposite of what model predicted (a faster RRP). Some explanation and discussion will be helpful.

The RRP refill rate indeed influences the transience in the mentioned way. However, its influence already starts earlier and is also influencing the overall amplitude (if some minimal background activation is assumed). It is therefore especially influencing the sustained component. However, for the nasal model already the inferred RRP size is the smallest and it seems that a small RRP refill rate is sufficient to produce the sustained response behaviour which we see in Figure 4f.

We thank the reviewer for this thoughtful comment and mentioned this behaviour in the discussion.

17. Also, the model simplifies vesicle transition rates by removing their calcium dependence. The Methods section indicates that this choice resulted from early fitting results that essentially "dialed out" the calcium dependence. Given the relative freedom that the model seems to have in finding suitable solutions, how is the lack of calcium dependence justified, and what potential impact might it have on the modeling results?

Identifying model (mis-)specification is a non-trivial task in general. The presented model is complex enough to replicate the recorded data but can easily be extended to more complex dynamics (e.g. more complex calcium handling) in future studies, as it is publicly available online. Further added components could even act as “distractors” to compare the other parameters across zones and we thus decided to use an “as simple as possible” model.

Interestingly our previous study (Schröder et al., 2019, Approximate bayesian inference for a mechanistic model of vesicle release at a ribbon synapse, NeurIPS.) showed that even at a temporal resolution of single released glutamate vesicles, it was not necessary to include calcium dependency for the refilling of the vesicle pools. This study thus supports our model choice.

18. Lines 503-508, "In combination with the approximately equal and opposite effects of calcium baseline on the detectability of On- and Off-events (Figure 7b,f), this suggest(s) that the calcium baseline may present a key variable that enables ribbons to trade-off the transmission of high frequency stimuli against providing an approximately balanced On- and Off- response behaviour." – what will be the physiological relevance for such conditions, perhaps the level of adaptation? Any existing data or predictions?

The reviewer raises an interesting but ultimately perhaps unanswerable point, given the scarcity of available data on temporal natural image statistics in the UV band across the larval zebrafish visual field. It is of course tempting to speculate that the ecological need to tune kinetics and On/Off preferences might be linked (e.g. detecting a “dark looming predator” might disproportionately benefit from a rapid Off response). However, to truly understand this idea at a useful level of detail would likely be a rather involved study in its own right. Accordingly, we here prefer to simply point at the possibility to “tune” the ribbon using calcium baseline, and what effects this might have on kinetics if all else was kept equal.

19. I am slightly skeptical of the predictions that the model might make about the ribbon's frequency tuning (Figure 7) in light of the fact that the AZ model in particular seems unable to reliably capture the fast transient response to dark flashes (Figure 4c,f).

The noted effect in the fast transient components in Figure 4c,f is partially due to the slow calcium recordings which act as an input for the model in Figure 4. As mentioned, and discussed above, there is an ongoing discussion to what extent nanodomain or more global calcium concentration drives the release. For this reason, we added a simple calcium model for the simulations for Figure 7 which includes a variable time constant for calcium (nanodomains would presumably have much faster calcium transients than used for the model default). This allows us to explore the influence of different possible calcium handlings. Although this extrapolation to new stimuli is based on the fitted model, it allows for varying all essential parameters. In the online simulation it can be observed that for fast calcium handlings the ribbon is able to also follow higher frequency stimuli. However, we agree that experimentally testing the influence of different ribbon configurations on frequency tuning is an interesting research direction but goes beyond the scope of this manuscript.